# Conceptual qualitative system dynamics model for simulation of perceived workload, stress and performance from industrial work content

Tuan-anh Tran[1,2], Tamás Ruppert [1,2*], György Eigner[3,4], János Abonyi[1,5]

1 HUN-REN-PE Complex Systems Monitoring Research Group, University of Pannonia, Veszprem, Hungary, 2 Department of System Engineering, University of Pannonia, Veszprem, Hungary, 3 Physiological Controls Research Center, University Research and Innovation Center, Obuda University, Budapest, Hungary, 4 Biomatics and Applied Artificial Intelligence Institution, John von Neumann Faculty of Informatics, Obuda University, Budapest, Hungary, 5 Department of Process Engineering, University of Pannonia, Veszprem, Hungary

* ruppert.tamas@mk.uni-pannon.hu

## Abstract

Workload and psychological stress, which industrial workers perceived as stressors, affected their performance after they were exposed to the work content. A model simulating the stress-performance of a working individual is a beneficial tool for work task design and production management, enabling long-term Human Resource Development. Current models and concepts lack the construction of work-content components, human centricity, and the mechanism of stress transformation and effect; therefore not able to reproduce subtle human behaviors. This paper formulates this problem with a multi-disciplinary literature review, and proposes a conceptual qualitative system dynamics model to simulate the stress and performance of workers in a given work environment and conditions. By replicating the changes in work content with associated effects, human-centric solutions and interventions can be designed. A use case in the Vensim environment with different simulated scenarios returned behavior and tendency in the outputs that aligned with phenomena reported in relevant studies. The model enables analysis of human factors in complex manufacturing systems, especially the effect of work content on individual workload-stress perception, benefiting future development of Human Digital Twins. This research calls for experiments and clinical trials to strengthen the existing associations between model factors and more effort for developing realistic mechanisms for modeling human factors in Industry 5.0.

## Introduction

Industry 5.0 (I5.0) suggested by the European Union (EU) embraces the well-being of human workers/operators [1] (hereinafter, the word "worker" is used for both terms). Within industrial manufacturing systems, workers can perceive psychological

**Data availability statement:** All data are in the manuscript.

**Funding:** Gy. Eigner was supported by the distinguished Professor program of Obuda University. The work of T. Ruppert is supported by the János Bolyai Research Scholarship of the Hungarian Academy of Science.

**Competing interests:** The authors have declared that no competing interests exist.

stress (hereinafter referred to as "stress") from three main sources [2]: the physical environment [3], the work context (i.e., the work settings), and the work content (i.e., the demand of assigned tasks [4,5]). As the first two sources have stable effects, the work content is the dominant source directly related to physical symptoms [6]. When workers face physically or psychologically demanding tasks that exceed their abilities or resources, the perceived workload becomes a stressor [7–10], causing physiological stress as mental or physical tension [4,11]. Too much work content causes high perceived workload, stress, and fatigue that negatively impact productivity performance and health outcome [9,12–16]. With the development of wearable devices and the integration of neurophysiological techniques, workload and stress status can be monitored and detected in real-time with physiological parameters [17–20], assessing cognitive load in real-world operational settings [21]. An ideal application is a platform for real-time monitoring of workload and stress [22,23], thus interfering with the work content adjustment, or Just-in-the-moment Adaptive Interventions (JITAI) when the perceived workload has accumulated to an unfavorable level [24,25]. Besides the current approach of stress recognition using data analysis on physiological parameters [26,27], a computational model reflecting the workload perception and the natural work-content-induced stress process can serve as a base simulation and prediction tool for enhanced stress recognition accuracy, paving the way for work-content design and adjustment as interventions, thus optimizing the performance. However, as current stress modeling efforts are not developed with consideration for this specific purpose, several aspects need to be tackled and elaborated.

Firstly, the configuration of work-content factors as stressors should be established, with corresponding effects and behaviors from different stress types. There are separate factors of the task demand, or elements of the work content that should be defined, such as physical and mental demand, time pressure, work pacing, and psychomotor factor [11]. While research studies often focused only on physical environmental stress [3], physical [28,29], or mental workload [30], industrial workers face a complex combination and interactive effect of them [31]. Given the fact that different levels of a factor can pose different interactions with another factor (i.e., physical workload on mental workload) and ultimately the performance [32], a model with a detailed configuration of work-content factors will support later design and modification of these factors. Unfortunately, current stress-performance models do not favor this approach. The stressors in these models either covered factors other than work content (e.g., organizational culture, interpersonal relationships with supervisors and coworkers [33], environmental factors such as heat stress [34]) or only focused on modeling the effect from one factor (e.g., mental tasks in an Information Technology (IT) company [35]). The work content either was neglected [36] or cannot be used in an industrial context (e.g., workload as academic credit hours, hours spent per week in sports [37]).

Secondly, as occupational stress is a multidimensional phenomenon reflected in physiological and psychological responses from an individual to a particular work situation [38], both the task and the individual factors should be considered in determining the desired regions of workload [11,39,40]. Personality difference, stress

reception, and work experience influence the perceived stress and response [41–43]. Therefore, a personal stress profile with adjustable parameters can offer a higher customization level for each interested individual, thus facilitating deeper construction of the stress exposure, later stress monitoring, and work-content effect study. However, a personal configuration was often neglected in modeling efforts [44].

Thirdly, since more than one stress type can occur during the work duration, the accumulation and transition between them should be formulated, with corresponding effects on performance. Acute stress in general, Acute Work-Content-Related Stress (AWCRS) [45] in particular, is characterized by short duration and engaged Sympathetic-Adrenal-Medullary (SAM) axis [46], with either positive or negative effects [47]. A low level of acute stress is associated with vigilant or sustained attention [48,49], improves decision-making and alertness, stimulates cognitive functioning, and cognitive capacity [50]. Consequently, this positive effect produces the optimal performance [51] with enhanced mental and physical efficiency [52]. Prolonged exposure under acute stress accumulates the allostatic load, exacerbates pathophysiology, causes chronic stress with long-term anxiety, psychological disorders, and increased risk preference [53–57]. This negative effect is associated with a decline in cognitive and physical functioning [58]. To utilize the arousal state of acute stress, besides a well-designed work content and environmental setup [59,60], a mathematical model to compute, simulate, and predict the stress level and performance with the intended work content in a risk-free manner before implementation can be useful and safe for stakeholders [61]. However, the configuration and transition of stress types are not well-formulated in the previous models. Although several suggested models [62,63] were based on the Yerkes-Dodson stress-performance curve [64], this arousal theory only considers the negative stress connotation and lacks considerations about the under-load condition or personal background [65]. Some models suffer from an underdeveloped categorization for AWCRS [66–68], and an ambiguous reciprocal effect between numerous stress types [69,70].

Fourthly, for the desired application of real-time monitoring, the stress modeling mechanism should reflect the reception of arriving tasks as discrete events in the time domain, leading to incremental stress accumulation and transformation within working hours. Other stress modeling efforts with Structural Equation Modeling (SEM) [71,72] are not suitable for this purpose, since they diagnosed the aggregated effect of static stressors after a working session. Though Benthem et. al. adopted a simple personal profile with cortisol half-life value and the modeling of work task reception as impulses in random moments facilitates the analysis over the time axis [73], this model structure does not consider the configuration of work-content stressors.

Last but not least, the final output of the modeling effort should aim at predicting the human performance under stress effects. Some computational models deliver output as a cumulative amount of cortisol, thus allostatic load, and then the disease status [73], but performance is not mentioned.

Recently developed models for a similar purpose of reflecting the workload-stress-performance relationship have not incorporated altogether these aspects in the model structure and simulation mechanism, with at least one aspect left untouched. Dear et. al. [34] modeled productivity loss in percentage, but only heat stress was considered, without discrete value calculation. The agent-based simulation model in Ref. [35] employed a stress level calculation and predicted productivity with task objects coming in time steps, but the tasks were only relevant to an IT office, with a lack of physical demand. Similar incompatibility can be found with the discrete-event stress-performance simulation of financial document processing tasks [61]. The human performance in the automotive line in Ref. [36] is calculated with a time step, but the work content definition is simplified, without a personal profile. There are dynamic models and platforms [44,74,75] aimed at simultaneous stress and attention monitoring for individual [76–78], to predict potential stress, tiredness, comfort level [75,79,80]; however, due to a lack of elaborated stress mechanisms and interdisciplinary approach [81], they are not able to simulate the stress effect from combination of work-content factors, which is critical for the design and assessment of industrial work content and stress-relieving interventions.

Motivated by the above-mentioned needs, this paper proposes a qualitative system dynamics conceptual model regarding the stress and the performance of an individual worker under the effect of the work content in a certain work

 

environment and setting. The system dynamics approach is chosen as the modeling technique since it can reflect and assess multiple non-linear behaviors and multi-loop structures over time [82] within a complex system of a human being (e.g., with physical, mental, and psychological behaviors modeled as internal sub-systems) while considering its interactions with the work environment and work requirement as external sub-systems. During model development, different stress behaviors and the effect of relevant work-content factors, and personal profiles with basic workload preferences were considered, which enables task design and stress profile customization. In return, the model distinguishes static and dynamic effects that reflect subtle stress-performance associations under different work scenarios. Based on the work content configuration and stress mechanism, stress-relieving interventions were suggested along with different demonstrated usage in the use case. This initiative not only aligns with the International Organization for Standardization (ISO) standard of developing an Occupational Health and Safety (OHS) management system [83] but also grants real-time support and maintenance of human performance level, which facilitates both I5.0 and Society 5.0 objectives [84] with daily action [85]. The proposed human-centric simulation model can significantly contribute to the field of industrial engineering in multiple ways:

- Supports the engineers in planning the optimal work content and work design for their workforce, considering the individual work capability and certain work environment;

- Provides the supervisors a base of expected behavior from their workers in the forthcoming work shift with a given work content, facilitating real-time stress-performance monitoring;

- Aids the manager in planning and adopting stress-relieving interventions, improving the turnover of human resources;

- Informs the industrial workers about how their work capability influences their perceived workload and stress, increasing awareness of well-being within the factory;

- Offers researchers and practitioners research tools, guiding their research and observation about the complex relationship between work content, work capabilities, stress, and performance;

The authors also urge more experiments or clinical trials that validate the effect of elemental work-content factors on individuals, to prepare for an application applicable to a large-scale workforce, building socially sustainable factories [86] while preparing for the Operator 5.0 (O5.0) paradigm [87,88].

## Problem formulation and preliminaries

This section describes the building pillars of the model with relevant preliminary theories, stress behaviors, and effects collected from the literature. The "workload" from industrial tasks as the "primary stressor" is considered in Section: Primary stressor: Task load – Workload, while the "personal perception profile" of the worker is discussed in Section: Personal perception: Personal profile – Personal capacity – Basic task load, with the "circumstantial factors" as the "secondary stressor" analyzed in Section: Secondary stressor: Circumstantial stressor – Stress exposure. The stress mechanism is elaborated in Section Stress mechanism, while Section Stress-induced states – Intervention – Performance suggests relevant integrated stress states, interventions, and associations with personal performance. A glossary of terms that are used frequently in the model construction is provided in the S1 Table of the Section: Supporting information.

### Primary stressor: Task load – Workload

As work content is considered the main source, the primary stressor in this model includes the task requirement that the worker needs to perform in a predefined work position. This subsection defines the elementary "task load" components (physical and mental) with the scope for each type (Section: Physical and mental task load), distinguishes between the "task load" and perceived "workload" (Section: Task load and workload), and defines the "workload component interaction" (Section: Interacted load).

**Physical and mental task load.** Although stress is a mental state, physical "task load" (energy, muscle, and physical strain) also affects mental "task load" (cognitive activities) [89] and contributes significantly to stress formation. The proposed model considers both these types as primary dynamic stressors:

- The physical "task load" with three components: posture, force, and time as inspired by Berlin et al. [90], and can be measured by separate measures such as REEDCO Posture Score Sheet [91] for posture, force in Newton and time in seconds, or a collective measure such as Cardiovascular Load (CVL) [92].

- The mental "task load" with four components according to the VACP model [93]: visual, auditory, cognitive, and psychomotor. This load can be measured by subjective methods such as self-reported questionnaires, i.e., Borg Workload Scale and National Aeronautics and Space Administration Task Load Index (NASA-TLX) [94].

The scope of physical and mental "task load", along with the proposed components, is explained in S2 Table of the Section: Supporting information, along with associated effects as stressors.

**Task load and workload.** While "task load" is the task requirement designed by production/industrial engineers, the "workload" is the perceived load that is subjective and dependent on individuals [95]. For ease of quantization, the authors proposed that each task should be designed with a known level of the seven above-mentioned load components, and the "workload" should also be decomposed into similar components. From existing literature, these "task load" components can trigger psychological or physiological stress on the operator in different ways: one "task load" causes many "workload" components, or many "task load" components have the same effect on one "workload" component. These effects can either be individual or interact with each other. Bad posture "task load" can cause posture "workload" (e.g., pain [96]) and cognitive "workload" (e.g., tiredness [90]). High force "task load" causes force "workload" (e.g., degraded force capacity [97]) and psychomotor "workload" (e.g., muscle fatigue [98]). High time and posture "task load" (e.g., fast work pace [99] with repetitive motions [100]) causes posture "workload" (e.g., musculoskeletal disorders [97,101]). Demanding visual "task load" [102] increases the visual "workload" and degrades the task performance. High cognitive "task load" requires high force "workload" (e.g., greater muscle activity [103]) and higher time and psychomotor "workload" (e.g., longer finish time and reduced performance [104]).

However, for the sake of simplicity, the authors do not incorporate all these possible individual or combined stress effects into the proposed model, but consider that each type of "task load" will exert an effect on the corresponding "workload" component. One approach to model and diagnose the effect of a "task load" component is quantifying the corresponding "workload" using multiple resources (visual, auditory, cognitive, and psychomotor), with an additive value at the beginning and being subtracted at the end of the task duration [105]. However, this technique is insufficient to reflect the fact that one person can feel an elevated workload when tired, or under unfavorable working conditions (i.e., noise, heat [106], being close to a robot [107]), or workers with more experience perceive a lower "workload" [108] for the same task. Therefore, each "workload" component in this model is introduced as a value that is dependent on each receiving personal profile.

**Interacted load.** Not only does the "task load" have a complicated effect on "workload", but the induced "workload" components can interact with each other, based on their amplitude, occurrence time, and duration [89], thus creating an additional "interacted load". This can happen within a type of "workload" as suggested by the cube model for physical "workload" [109], e.g., a combination of poor posture and fast repetitive task [110], or due to the effect of one type on another, such as demanding physical "workload" leads to decreasing situational awareness [111] and higher mental "workload" [32], while physical capacity (regarding fatigability and recovery) is negatively affected by mental "workload" [112]. For the sake of simplicity, this study defines "interacted load" as an additional amount of "workload" on the most prominent component, which occurs if some "workload" components exceed a predefined value. For example, the interaction between posture "workload" (i.e., overhead activities) and time "workload" (i.e., fast repetitive task) places

additional workload on the force "workload" (i.e., from the shoulder muscle) in the study of Ebaugh et al. [110]. This simplification is configurable, depending on the simulated purposes and the environment.

Those mentioned above physical and mental workload components will be considered throughout the combination of "task load" with "personal perception" that creates the "perceived workload" as illustrated in Fig 1. Each "task load" has three physical and four mental components. "Workload" components can pose additional "interacted load", thus the "work-load" is the total combination of "perceived workload" and "interacted load".

### Personal perception: Personal profile – Personal capacity – Basic task load

This subsection describes how each worker perceives a "workload" from a "task load" differently, based on the "personal profile" (Section: Personal profile) and "personal capacity" (Section: Personal capacity), with the "basic task load" is defined as work preference (Section: Basic task load).

**Personal profile.** Each worker has a unique "personal profile" from professional and occupational backgrounds, setting up initial conditions before a working day, and how a stressful work demand is received. An adjustable set of factors is proposed for this purpose, which are characterized by long effect periods (months, years) and can be categorized into different groups:

- Static profile: Factors that require time to undergo a natural increment or degradation without external intervention, and can be considered static, e.g., work experience, age, physical impairment, and chronic stress effect.

- Dynamic profile: Factors that have long effect periods but are subjected to change under possible training and intervention during the working session, such as training experience, skill decay, and problem-solving ability.

- Stress-related profile: Factors that are explicitly related to the stress accumulation mechanism of a person, such as stress endurance, thresholds for task demand and capability, sustained attention value and duration, acute stress value and duration.

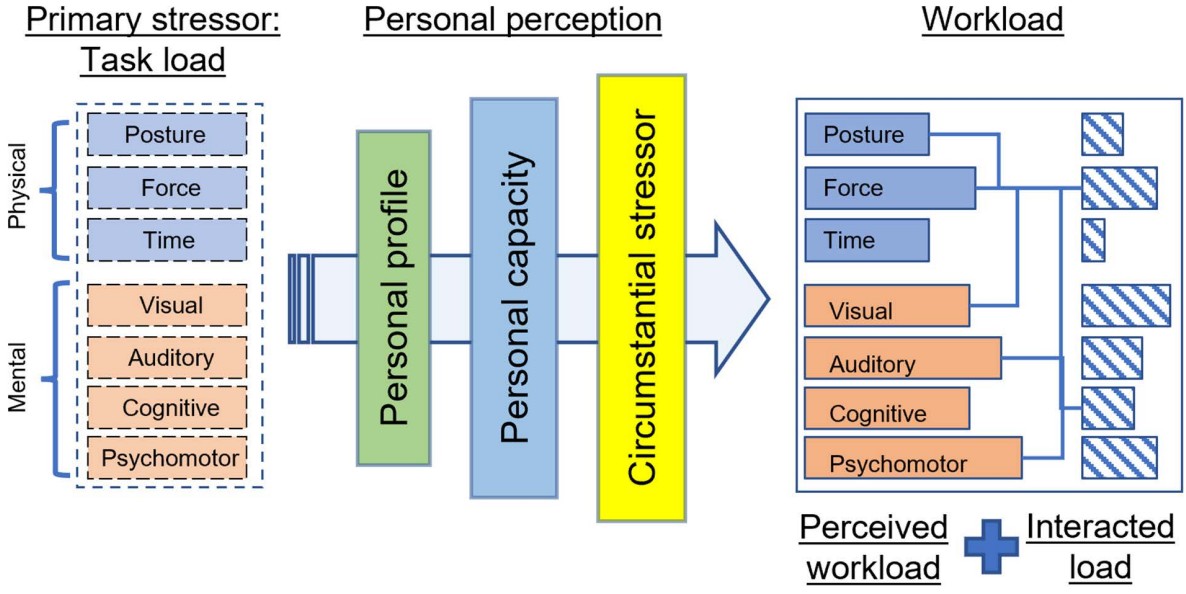

**Fig 1. The relationship between "task load", "workload", and "personal perception".**

More details of the considered factors in the "personal profile" can be found in S3 Table of the Section: Supporting information.

**Personal capacity.**  Work capacity was widely analyzed in clinical research [113] as the physical and mental capacity for work. Each individual with a certain level of physical fitness has a predefined Physical Work Capacity (PWC) [114], representing the available energy [115]. Mental capacity can be determined in a similar perspective [116,117]. While physical work capacity is affected by personal experience, training, motivation, and environmental factors [118], mental capacity is influenced by innate characteristics [119], historical medical record [120], and both capacity types are affected by common factors such as age [121,122].

This research suggests each worker has a "personal capacity" with six components corresponding to "task load", namely posture, force, visual, auditory, cognitive, and psychomotor, except for the "time" load. Inspired by the concept of workload margin [123] and maximal mental capacity [116], these capacities behave as resources with a "natural degradation" over time, while further reduction happens when the worker experiences the negative effect of stress (e.g., encountering a heavy task load or complex problem), which is named "stress degradation". An ideal worker with normal physical and mental conditions (i.e., no physical impairment) has 100 percent of each "work capacity" at the beginning of the work session. Each "personal profile" sets up a different "initial personal capacity" that is less than this optimal value, representing their competence for work. For example, a worker who has a minor injury in one arm can start the working session with 90% of posture capacity and 80% of force capacity, while having 100% of all mental capacities. The "Personal capacity" of a worker when receiving a task indicates the actual capacity at that time, making the same "task load" can be perceived differently [124]. Except for time, six other "personal capacity" components follow the same behavior illustrated in Fig 2. When the work began on the first milestone, the worker hardly felt any deviation between "task load" and "workload". Under the "natural degradation" through time toward the sixth milestone, the difference becomes significant. Fatigue happens at the fifth milestone when the worker perceives a task as demanding and over the current "personal capacity", though the "task load" remains the same. The "basic task load" can generate a "workload" that does not exceed the "personal capacity" at the end of the work shift. With the "capacity degradation" equaling the total of "natural degradation" and "stress degradation", fatigue can come earlier (i.e., at the fourth milestone), and even the "basic task load" can pose an

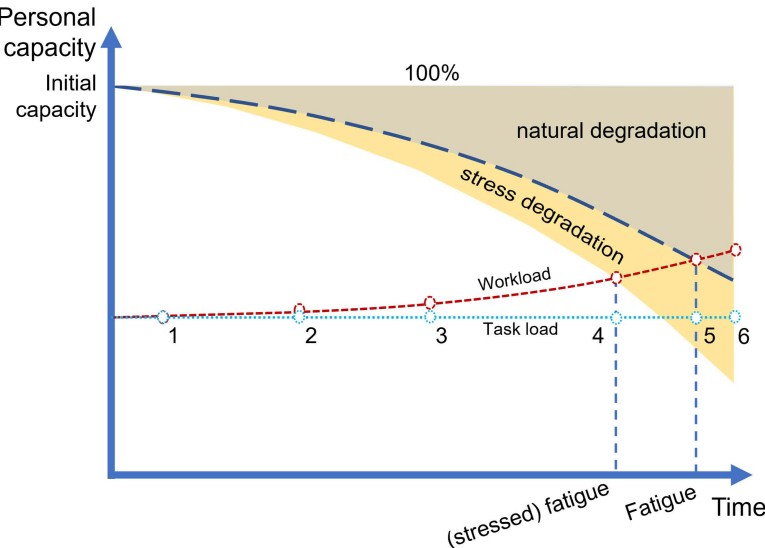

**Fig 2.  The relationship between "task load", "workload", "basic task load", and "personal capacity".**

overload status at the end of the working session. In contrast, it is assumed that a "motivated capacity" (not shown in the Figure) can be achieved as an arousal state, under which the worker has a vigilant state and slower capacity decreases.

Instead of a "personal capacity" for the time factor, a worker will have a "time variation" which reflects the work proficiency, and this variation will be reduced by the learning effect [125]. New workers during the skill decay period may have higher "time variation" than old and experienced ones [126]. As the available "time" in a work shift is equal for every worker, the "time" factor does not follow the "natural degradation". However, time already affects the accumulation and relaxation of all stress types, while the occurrence of "time pressure" increases the "perceived workload" [127], defect/problem probability [128], and "capacity degradation" of the worker. Though the worker perceived the time "task load" as an indirect factor, a personal preference for a working pace still exists, which can be considered as the "basic time load", as mentioned in the next subsection.

**Basic task load.** Considering each worker has a limited PWC, the physical "basic task load" is the safety margin to work on a specific task with no sign of fatigue throughout a work session [129]. The "basic task load" can be identified with physiological indicators such as the Rating of Perceived Exertion (RPE) and relative Heart Rate (HR) [130], for a predefined duration (i.e., 4-hours, 8-hours, etc.) based on the Maximum Acceptable Work Time (MAWT) [131]. The mental "basic task load" can be defined similarly [116].

This research proposed seven "basic task load" components that correspond to the seven "task load" components and should be measured and set up individually from the beginning of the work session and validated throughout a certain period. Any loads that exceed these values will pose a demanding situation for the workers. The "basic time load" has two components: "basic task time" (i.e., the duration required to finish a task) and "basic pace time" (the interval between two adjacent tasks). Any deviation from the basic values can create "time pressure" [108], which imposes an additional demand and perceived strain on the worker [132]. The relationship between the "basic time load" components and the time effect in the model is illustrated in Fig 3. If the "time variation" of the worker varied within the "no perceived pressure" region around the "basic task time", there is no effect of time load on the perceived workload (e.g., task 1 and 2). This is the same situation when the worker can complete a task in less than the basic time load (task 3), when the task is too easy, the allowed time is longer than needed, or the worker has a higher skill than expected. In these tasks, no pressure is perceived, no stress is accumulated, and the stress "relaxation rate" is increased. The effect of the "basic pace time" is similar: for tasks 1, 2, and 3 that come at a normal pace, there will be no "time pressure". Between tasks 3 and 4, the worker has time to recover from stress. However, if one task is missed due to a problem or rework (e.g., task 5), then the worker will face "time pressure" when the next task comes, similar to the situation of two tasks coming in a short period (tasks 6 and 7).

## Secondary stressor: Circumstantial stressor – Stress exposure

This subsection discussed environmental and work setting factors as "circumstantial stressors" (Section: Circumstantial stressors), along with their exposure effect (Section: Circumstantial stressors).

**Circumstantial stressors.** Other factors that are related to the work environment and setup (e.g., buffer level [133], the physical environment [134,135]) also determine how the worker perceives the primary stressors from the "task load", thus posing an additional "workload". Therefore, these "circumstantial stressors" are considered secondary stressors and can be categorized based on their temporal behavior:

- Static stressors: Factors depending on the surroundings, work setting, and initial setup, such as environmental disturbance and buffer level. These factors are set up at the shift beginning and remain unchanged throughout the working session; therefore can be considered static.

- Dynamic stressors: Factors that heavily depend on the natural characteristics of the work and have varying values throughout the working session, such as working hours, failure rate, and work pace, are also dynamic stressors.

 

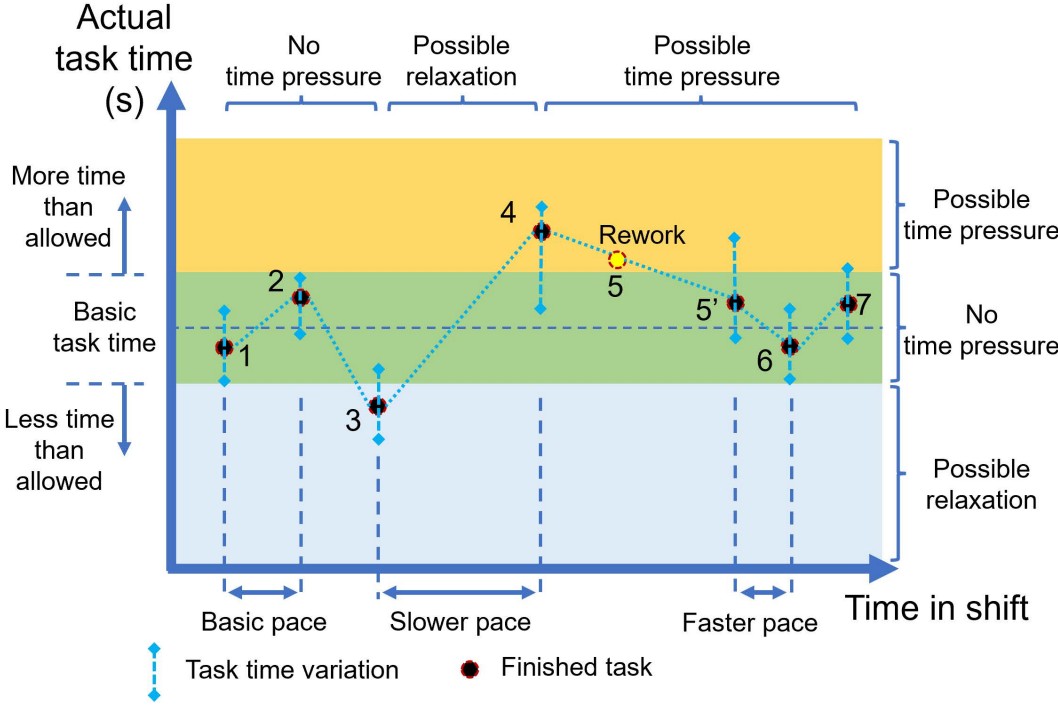

**Fig 3. The relationship between basic time load and time pressure.**

**Circumstantial stress exposure.** When an individual with a "personal profile" is exposed to a "circumstantial stressor" (i.e., being assigned to a certain workstation (WS), with a certain physical setup), two types of effects can be distinguished:

- Static effect: The "initial personal capacity" of a worker is affected by the natural characteristics of the assigned WS and its associated tasks, in both positive and negative directions. Unfavorable setups (i.e., poor lighting, unergonomic design, heat [136], noise [137]) cause a capacity reduction or additional load [137], or vice versa, additional buffer quantity gives more time for responding, which increases the "basic work pace". These effects are pre-determined at the beginning and consistent throughout the work session.

- Dynamic effect: Stressors such as occurring problems and machine failures randomly occur, and pose an additional "workload" and contribute to a faster "capacity degradation" than the "natural degradation" (e.g., poor lighting reduces visual capacity significantly).

Several "circumstantial stressors" and their effects from the literature can be found in S4 Table of the Section: Supporting information. Fig 4 provides an example describing the effect on an individual worker (i.e., worker "A") from the static "circumstantial stressors" of an imaginary workstation (WS) of a production line, such as unergonomic design, a certain Work-In-Process (WIP) buffer, and poor lighting, or with dynamic ones such as material defect and machine breakdown. Due to the unergonomic cell design, the posture, force, and psychomotor capacities are reduced further than the "initial personal capacity", with 10%, 15%, and 5% respectively. Due to performing uncomfortable movements, the posture capacity degrades 15% faster than the "natural degradation". Due to the low quality of input materials, worker "A" therefore feels an additional visual and cognitive load to check the incoming materials carefully. Thanks to the high number of WIP buffers, the worker has an additional 10% of capacity to cope with the time requirement of the task. Due to the poor

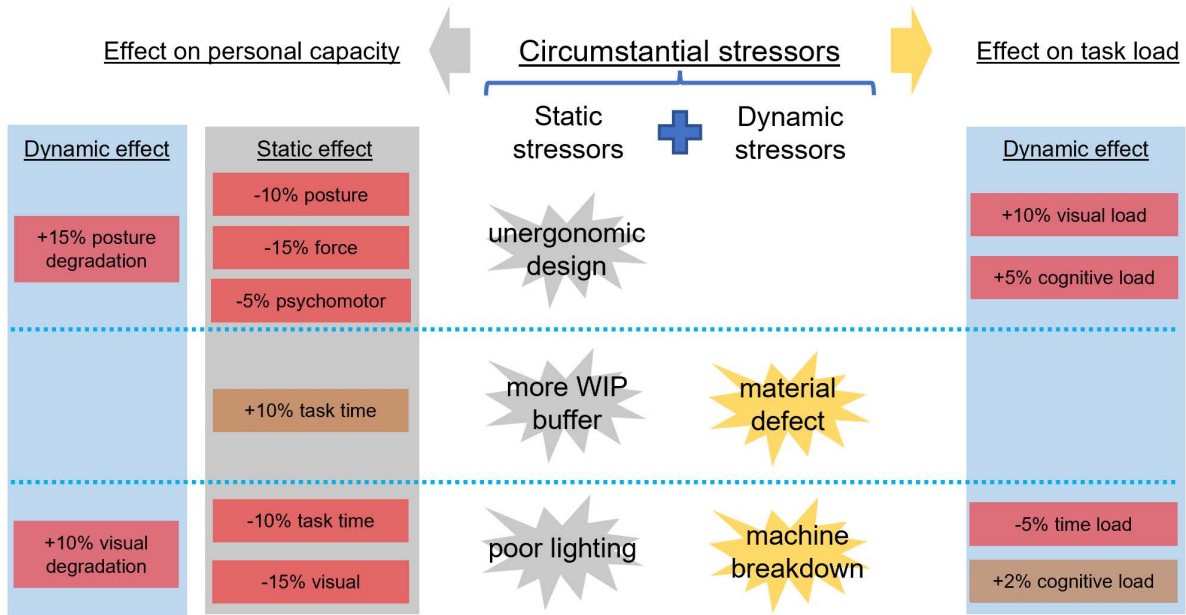

**Fig 4. The static and dynamic effect when worker "A" is exposed to circumstantial stressors.**

lighting condition, worker "A" loses 10% from the basic task time as more time is needed to react and recognize problems, and loses 15% of the visual capacity, while the visual capacity degrades at a rate of 110% than the "natural degradation". Frequent machine breakdowns in this WS create time and cognitive loads to adjust the machines, which makes worker "A" feel more "workload". The effect of these stressors on the "task load" and "personal capacity" can be set as linear or non-linear, additive or multiplicative, dependent on the intrinsic characteristics of the industrial tasks of interest and the context of the manufacturing environment.

## Stress mechanism

This subsection describes the "workload reception" process when the worker encounters a task (Section: Workload reception) and the stress mechanism that happens afterward (Section: Stress accumulation, recovery, and transformation).

**Workload reception.** Due to the complex nature of job strain/stress perception, the authors proposed a simplified mechanism for workload perception based on the Demand-Control model of Job stress by Karasek [138], by comparing each "workload" component with the corresponding capacity of the operator, and then aggregating all the perceived differences under the category of "perceived situational demand". The operator then perceives stress when this aggregated demand is higher than the handling capacity, i.e., higher demand than control. In the working process, under an ongoing task, two aspects indicate if a worker experiences a demanding situation:

- Perceived situational demand: The difference between the incoming "task load" and the "basic task load" is the "perceived situational demand". The more surplus value of task components, or the more task components that exceed the "basic task load", the more challenging the incoming task is. The average value of "perceived situational demand" is calculated from these component differences, considering the current "personal capacity" at the time of receiving the task. If the average overload exceeds a certain personal threshold for "Demand", the worker can feel this difference and perceive the task as too demanding from a physical or mental aspect.

- Perceived capability: The "perceived capability" is achieved similarly by comparing the current "capacity degradation" with the "natural degradation". If the stressors cause the "capacity degradation" to be drastically reduced compared to the natural rate, it can create the feeling of lacking the required capability to perform the work. The average value of "perceived capability" is calculated from the component differences regarding the current "personal capacity", and if it exceeds a personal threshold for "Capability", the worker perceives himself as incapable of handling the incoming "task load".

The current "personal capacity" of the worker plays an intermediate role in calculating both of the above-mentioned average values. Lower capacity levels show a tiredness and exhaustion status, thus the worker feels the load more demanding and sees himself more vulnerable, even when facing the same level of "task load". Generally, when the "Demand" is greater, while the "Capability" is lower than the personal threshold, then the worker will consider the current task as a kind of threat and trigger the stress response mechanism. Fig 5 illustrates an example "task load" reception process. Assuming that the "basic task load" of a worker "A" is known, an incoming "task load" that has a posture score of 7 on the REEDCO scale and a visual score of 3.7 on the VACP scale is higher than the "basic task load", thus making the task challenging and "A" hardly perceives the other easy task components (e.g., the required force is only 20N, less than the preference of 50N). Only the challenging task components are considered for calculating the "Demand" score, regarding the elasticity of the current "personal capacity". The same relative comparison is between the actual "capacity degradation" and the "natural degradation". The posture, time, visual, and psychomotor capacities that degrade faster than

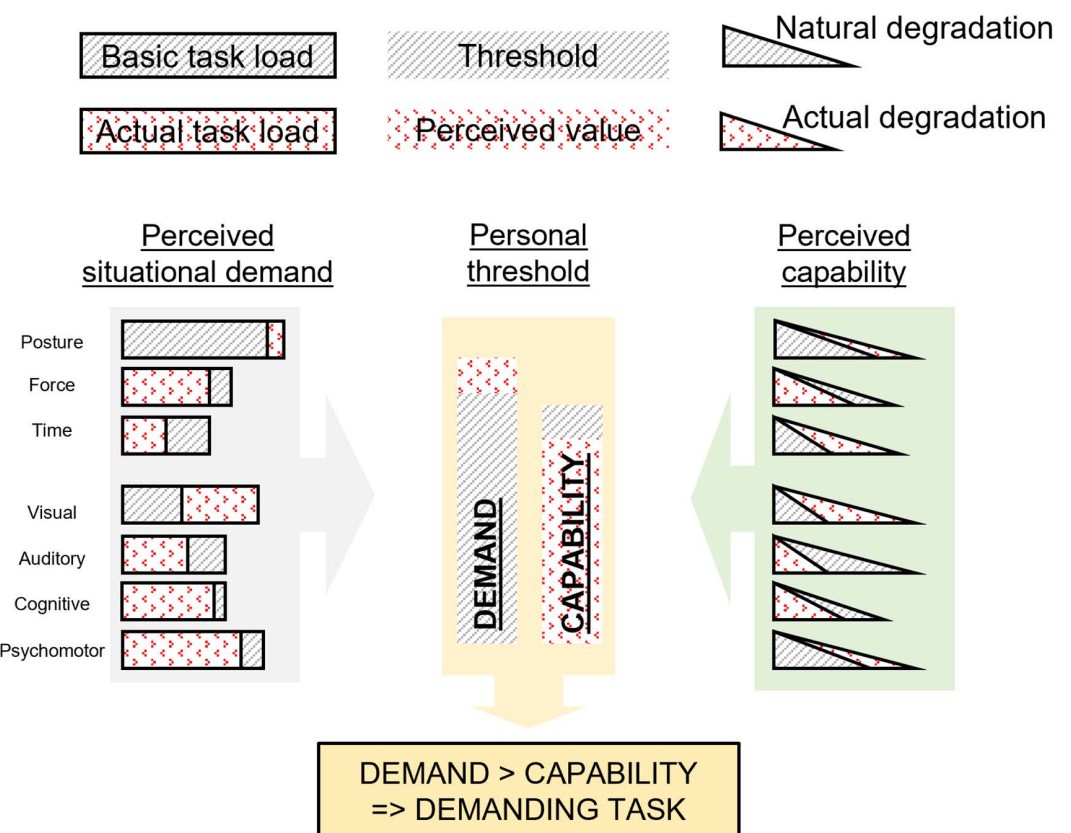

**Fig 5. Example about task reception of perceived situational demand and perceived capability of an imaginary worker.**

the normal rate affect the incompetent feelings and are used to calculate the "Capability" score. As the "Demand" score is higher and the "Capability" is lower than the threshold, worker "A" perceived this "task load" as demanding.

**Stress accumulation, recovery, and transformation.** The stress in this model undergoes the transition from the demanding task that the worker received, to the perceived attention, vigilance/"sustained attention" [139], "acute stress", and "chronic stress". Each type of stress has an accumulation and a recovery period before transforming into another type.

Monotonous and easy tasks result in disengagement and drowsiness, yielding negative individual and cooperative outcomes [140]. When perceiving a "demanding task", "sustained attention" is accumulated with an "accumulation rate", creating the positive vigilance attention [141] that breaks out the risk of "boredom" and ignites the "arousal" of the worker with an elevated corticosteroid stress hormone (cortisol in humans) [142] which enhances the working memory [143]. Under this condition, the worker experiences high awareness with "enhanced capacity" (i.e., the "capacity degradation" rate is reduced than the "natural degradation"), thus improving the efficiency [144]. "Sustained attention" has a "natural relaxation" rate that differs individually, which takes effect in the idle periods between going tasks and forms lapses when mind wandering can happen and the worker disengages from the current task pressure [145].

However, this positive effect will be reversed if the stressor exists for a long period [146]. If accumulated "sustained attention" accumulates over an "attention endurance" or intensity level [147], namely "value threshold" or "duration threshold", it becomes another source of stress [44,51], and "acute stress" is activated. In a normal healthy adult, these short-lived acute reactions decrease and disappear when the stressors cease [148,149], thus this short-term stress also has a "relaxation rate" activated if the perceived attention returns to a normal level. Similar to "sustained attention", "acute stress" is relaxed only when it is not accumulated, but at a slower rate. This recovery process takes place during work time (internal recovery). Once "acute stress" is in effect, the positive effect of "sustained attention" fades, and the worker feels the working capacities degrading faster than the "natural degradation".

"Chronic stress" accumulates with a similar mechanism: if the "acute stress" exceeds the personal threshold value and duration, then "chronic stress" appears, which brings long-term influences on the "personal profile" (i.e., reduced stress endurance), leaving the worker with less "initial personal capacity" before a new working day. The detrimental effects and relaxation of "chronic stress" are not considered in this model. The appearance of "chronic stress" comes along with "fatigue", while "burnout" is the extreme state with an over-arousal level. These stress-induced states and their effects on performance are discussed in the following.

Fig 6 illustrates an example of the stress mechanism, while each stress type has an accumulation and a recovery mechanism (i.e., a "natural relaxation" rate that takes effect in the idle periods between going tasks), before transforming into another type. When a worker perceives a "demanding task", "sustained attention" is accumulated with an "accumulation rate", which breaks out the risk of "boredom" and ignites the "arousal" of the worker with "enhanced capacity" and improved efficiency. When this very same worker faces demanding tasks repeatedly for a long time, based on the duration or the demand level of the incoming tasks, the physiological stress of this worker will undergo accumulation and transform into a higher level of stress. For example, if "sustained attention" accumulates over "attention endurance", "acute stress" is activated with negative effects. If the "acute stress" exceeds the personal threshold value and duration, then "chronic stress" appears, which brings long-term influences on the "personal profile", and takes a longer time to relax, therefore not included in this study.

## Stress-induced states – Intervention – Performance

This subsection describes different states that are incorporated into the proposed model (Section: Stress-induced states), how personal performance can be predicted from predefined parameters (Section: Performance profile), and different interventions can be considered (Section: Stress-relieving intervention).

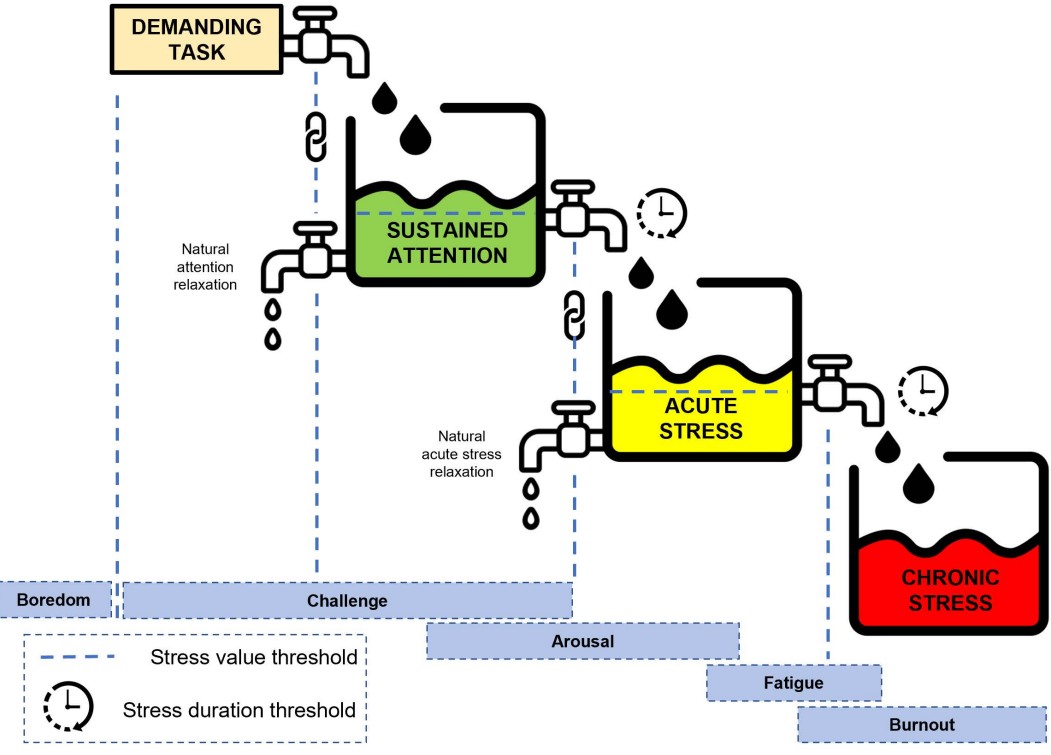

**Fig 6. The accumulation and transformation mechanism of different types of stress.**

**Stress-induced states.** Under the effect of work-content, the worker may experience different states with characterized symptoms and behaviors as follows:

- Under-load: Working with an easy, not challenging, and repetitive "task load" causes boredom and sleep transition [140,141,150], leads to human labor waste, occupational discomfort, and mental illness [151,152]. "Under-load" is considered as detrimental and stressful as "overload" [153], and prolonged exposure also leads to "fatigue" and injuries [154]. Interestingly, this condition is neglected in many relevant research [65]. In this model, this state is characterized by a repetitive pattern of "sustained attention" for a long duration without the appearance of demanding tasks.

- Optimal performance: A moderate "workload", under an acceptable "circumstantial stressor" exposure helps the worker escape the "under-load" condition with an aroused vigilance and engage with the current task [51], thus yielding the best performance [14]. This state can be recognized with the regular accumulation and relaxation of "sustained attention", possibly with a low and intermittent quantity of "acute stress".

- Overload: When the worker perceives an excessive "task load" exceeding the "personal capacity", stress is stimulated by both increased need for work capacity and current capacity decrease [150], negatively affecting job satisfaction, turnover intentions, and performance [155–157]. In this model, a continuous "acute stress" existence causes a faster "personal capacity" degradation.

- Fatigue: Working with any physical or mental "overload", or using a maximum capacity for a long duration leads to "fatigue" [158,159], which results in decreasing muscle performance in different body parts [160], cause the worker to

fail to maintain a required force, or feel tired and lack of energy [161], leads to reduced functional capacity and performance decrements [162]. This state can be identified when one component of the "personal capacity" is depleted.

- Burn-out: Under adverse working conditions with the existence of "chronic stress", the worker is burdened with seriously overwhelming exhaustion and impaired job functioning [163], therefore, the productivity and coping skills are significantly reduced [164] with evoked depression and negative attitude towards life [165]. This serious state is associated with the depletion of all "personal capacity" components; thus, the worker can quit the position even in the middle of the working session.

As formulated by Hebb [166] and Teigen [143], both "under-load" and "overload" states lead to lower performance, and a medium workload condition can induce an optimal level of response and learning, which results in "optimal performance" [51]. The next paragraph discusses how the "performance profile" can be analyzed based on the previously elaborated factors.

**Performance profile.** Inspired by the Lean philosophy, personal performance is assessed with three aspects of the Overall Labour Effectiveness (OLE) [167,168]: "availability-", "productivity-", and "quality performance". In other studies, "productivity performance" is measured as the number of completed orders per time unit [169], supervisor reports [163], or effective working time [170], while "quality performance" is measured by the number of correctly assembled parts [171]. However, given the stochastic nature of human behavior, this model only predicts these performance levels as the probability that a single worker can produce the appropriate output quantity and quality. The personal "performance profile" is assessed using three aspects of the Overall Labour Effectiveness (OLE), as suggested in Fig 7 within the framework of force field analysis. Availability performance is assessed by the capability of keeping a predefined work pace, which is supported by time and psychomotor capacities. If these capacities degrade faster than the "natural degradation" (due to stress), then the worker becomes less able to keep the standard pace. Availability performance is hindered by distraction possibilities such as work pace increment and problem occurrence rate. Productivity performance is affected by physical degradation (which is dependent on the force and posture capacities), and the physical degradation under stress (dependent on the degradation of force and posture capacities caused by stress). Quality performance is associated with quality-related attention (which is related to the degradation of cognitive and visual capacities, i.e.,

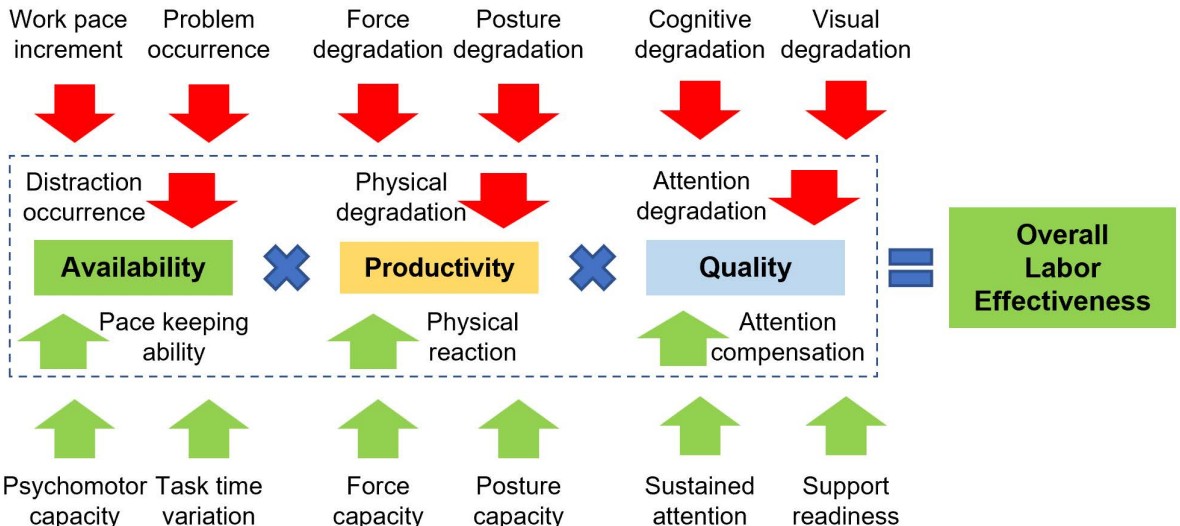

**Fig 7. The Overall Labor Effectiveness with relevant factors, in the format of drivers and restrainers in a Force Field Analysis.**

for tasks that require a visual check of input materials). As "sustained attention" can help the worker stay focused, the quality attention compensation also contributes to "quality performance". A general assumption is that at the beginning of the working session, a worker has a 100% probability for these three performances, according to the "initial personal capacity". These performances will vary according to the external influence factors during the work session. The "optimal performance" of a worker with maximum OLE can be reached when the predefined work pace can be kept up, maintaining good physical maneuver and focused attention. In the long term, the OLE still decays due to the natural depletion of the "personal capacity". The cooperation between workers is not considered in this model.

**Stress-relieving intervention.** The elaborated conceptual model is not only capable of reflecting personal performance under specific working conditions but can be used as a platform to deploy stress-relieving interventions, which are modifications or interfered activities that can be applied to change the stress process. From the stress structure and mechanism proposed in this model, possible interventions can be developed by the following principles:

- Refresh the static stress effect: By applying changes to the work environment and "circumstantial stressor", the stress profile can be refreshed.

- Easy the stress accumulation: By increasing the time interval between tasks, or reducing the "workload" to lower than the perceivable limit, the "sustained attention" can be disrupted, and the "natural relaxation" rate can be activated.

- Mitigate the "capacity degradation": By keeping the actual skill degradation as close to the natural one as possible, the worker can maintain the performance for a longer time.

- Increase stress resilience: By introducing a guided practice of stress management techniques, the worker can improve the personal "sustained attention" and "acute stress" thresholds. The coping and problem-solving capacities can also be improved through human resources management and training initiatives.

To ensure the successful integration of these interventions into the simulation environment, their scope and direction of effect should also be scrutinized as the integrated stressors in the previous subsections. There are two main categories of intervention: work-content modification as an intervention, or a non-work intervention. These interventions can be self-deployed by the worker, such as drinking coffee with a reasonable dose [172,173], and self-pacing breathing rhythms [174]; however, organizational guidance and suggestion play an important role in a successful implementation. Besides, some interventions can be intentionally implemented by the managers within an incorporated continuous improvement initiative, such as Lean manufacturing [133,175]. Popular interventions are designing workload variation in repetitive work [101], designing work-rest schemes allowing stress recovery [101,176], modifying workforce scheduling [170], changing the conveyor speed, rotating the job position to avoid boredom [28,175] and musculoskeletal disorder [177], providing a natural virtual environment during acute stress [178], ease the time pressure by controlling the display time [179], increasing workplace ergonomics to aid a certain workgroup [180], initiating physical exercise to prevent the deterioration of work ability [181].

Some interventions have a lagged effect or take time to shape such as giving rewards [182] or job motivation [183], increasing stress endurance [184], whose effect is hard to measure and control, therefore, only interventions with shorter effect duration and aiming at "sustained attention" and "acute stress" are considered. This research will not dive into the details of them.

## Human-centric stress-performance simulation

To validate the proposed concept, a qualitative system dynamics model was constructed in the Vensim simulation environment [185]. As explained in the previous section, the stress mechanism is built on the Demand-Control model of Job stress with the assessment of demand-control difference in time; therefore, its intrinsic characteristics are similar to the discrete-event approach, and will reflect the best scenarios with similar natures of intermittent coming tasks repetitively.

Thus, to simply exhibit the applicability of this approach, an example model will be constructed with characteristics of repetitive assembly work on an industrial conveyor, in the most general way that it can be configurable for other work. This section describes the industrial background with a predefined "personal profile" representing a worker of interest, to test the modeling capability of reflecting the above-mentioned working process, stress mechanism, and stress-induced statuses under different scenarios.

## Description of industrial assembly line environment

The indoor physical environment in nowadays industrial assembly lines is usually characterized by a certain lighting condition, temperature, humidity, noise [186,187], vibration, and associated ergonomic level [188,189]. As the effect of these environmental factors on human perception is mentioned in plenty of research, only the lighting and ergonomic factors will be considered in this model, assuming the illuminance is slightly lower than the recommended value of 300 lux for assembly work [190] and the ergonomic score is fairly above an ideal balanced posture [191] with 10% of body asymmetry. Though only individual performance will be considered without interaction with any colleague, it is assumed that social support is available in case of need, thus mitigating strains when they occur [192]. Other conditions (noise, humidity, etc.) can be set at normal conditions, with an imaginary failure rate of 205 seconds, which generates an additional "task load" when it occurs. To avoid the complexity of the accumulation of musculoskeletal symptoms [193], the current weekly working hours are 30 hours.

Inspired by the characteristics of assembly work mentioned in the literature [193–196], the position under simulation in this assembly line is characterized by different work-content requirements with a common feature of repetitiveness, which requires the design of the "task load" and a "work pace". For simplicity, the "task load" involves a fair posture score (5 on the REEDCO scale), a force of 10N, an average task time of 30 seconds, 90-second work pace, a visual quality check with a sample product, low requirement of auditory on checking the working tools, a fairly simple autonomous task and discrete actuation during assembly (scored 3, 3, 2.5 and 2 on the VACP scale, respectively).

The "personal profile" is created with different work preferences and stress-related profiles. Assumed that the worker is middle-aged with good experience, normal physical condition, good learning ability, in the stable phase of the skill decaying period, with intermediate problem-solving skills, and an average ability to cope with stress. For simplicity, this information will be translated as constant values in the model input, for example, middle-aged means the age effect is 1, while after the threshold of 45, the age effect will be degraded [197] to lower than 1. Normal physical condition and stable skill decay mean all these effects are not taking place, and the values are 1. These values are adjustable, depending on how the relevant curves or look-up functions are defined, customized to an individual of interest. Our worker has a "basic task load" slightly higher than the required "task load" (e.g., 25N to 10N, respectively), which means a working day can be carried out with no negative consequences. Extreme conditions such as sleepiness and serious physical disabilities are more complicated and require in-depth modification; therefore, excluded from this use case. Visualization of components of the use case and their relationship is shown in Fig 8. After defining the personal profile and work capacity of the worker of interest, the circumstantial stressors in the workstation and associated tasks are assessed. Based on these data, the stress-induced state and performance profile can be simulated.

To narrow down the scope of the use case, some assumptions are made as follows:

- At the shift beginning, the worker has a "personal capacity", which is affected by the "Personal profile".

- At the end of the shift, the remaining capacities in case of stress-free work can be estimated; therefore, the "natural degradation" can be determined.

- The "circumstantial stressors" of the assigned WS have a static effect on the worker, at a constant rate proportional to the uncomfortable level.

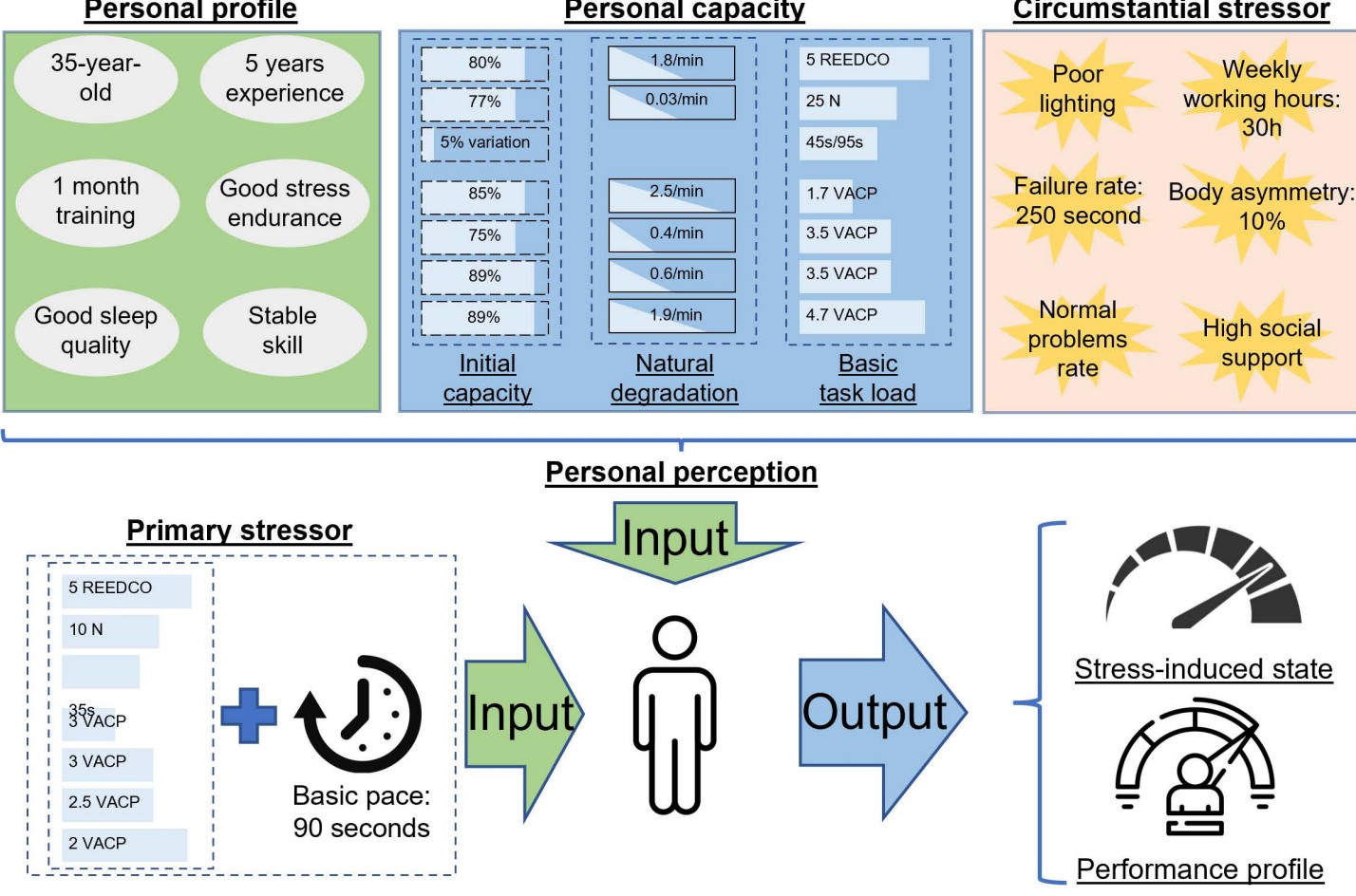

**Fig 8. The components of the use case, with the constructed "personal profile", "personal capacity", along with "primary stressor" and "circumstantial stressor" as inputs, while "stress-induced state" and "performance profile" are the outputs of the simulation.**

- When encountering incoming "task load" as stressors during work, the worker experiences a situational demand proportional to the perceived "workload".

- For simplicity, it is assumed that the components of the perceived "workload" are under their thresholds, thus there will be no "interacted load".

- The "accumulation rate" and "natural relaxation" of each stress type can be measured by the unit of "stress per second".

- Three types of stress have their effect in different confidence intervals of 15, 30, and 60 minutes for "Sustained attention", "Acute stress", and "Chronic stress", respectively.

- When the worker is under the effect of a stress type, the "stress degradation" is proportional to the current amount of that stress.

- At the shift beginning, the worker has a 100% probability of yielding the expected "availability-", "productivity-", and "quality performance", which can be estimated from the "basic task load". These performance constituents are naturally degrading but can be optimized or prolonged until the end of the predefined shift length.

The key elements of the model, such as stock and flow variables, causal relationships, and feedback loops, as well as how each element is measured and integrated into the simulation, will be explained in the next subsection.

## Model structure and simulated scenarios

This subsection focuses on explaining the example SD model constructed for the described use case, utilizing the conceptualization and mechanisms categorized in the previous theoretical section. During the model construction, the components of the model are constructed as variables, stocks, or flows with pre-defined principles as follows:

- Inputs, and set-up parameters for the personal profile, work environment, work scenarios, etc., are constructed as variables, which can be a fixed value or look-up value from a functional curve, as inspired by the model of affecting factors to cognitive load during dynamic learning [198]. The task load components are values generated randomly around the corresponding predefined expected benchmarks. These variables are considered as static background factors with only a few change points and do not have any accumulation effect.

- The "task generation" function contains factors that create the dynamic characteristics of the simulation, including a work schedule of 1 or 0, indicating that a work task is or is not carried out at a certain moment during the simulation time, respectively. During lunch break, there is no incoming task. There is another similar schedule indicating the occurrence of problems and failures. The multiplication of the working and problem occurrence schedule with the task load components represents the taxing task load on the worker at each time step of the simulation.

- Main parameters of interest, such as current levels of six elemental work capabilities (i.e., "posture workload", "force workload", etc.), current level of "sustained attention", "acute stress", and "chronic stress", are elaborated as stock. The stock values of six elemental work capabilities are always reduced due to the corresponding "natural degradation" and "stress degradation" as output flows at a constant rate. The effect of stress is considered additive and occurs only when the stress status is 1. For the stock values of stress types (i.e., "sustained attention", "acute stress", and "chronic stress"), these values are accumulating based on the working duration and the challenging level of the generated tasks as incoming flows, while the relaxation effect acts as outgoing flows and reduces the stock values. This accumulation mechanism is inspired by the working capacities modeled by Keller et al. [105].

- The effect and relationships of inputs and set-up parameters to other model components (i.e., variables and stocks) are reflected by links.

- In each time step of the simulation, the stress status is a binary variable that can be yes or no, in respective numerical values of 1 or 0, when the accumulated value of the corresponding stress stock exceeds a predefined "personal threshold" (e.g., "sustained attention threshold", "acute stress threshold").

- The elemental components of the OLE calculation are constructed as variables since they do not have this accumulation effect, similarly to how employee performance was modeled from workloads, skills, and motivation in Ref. [199], or how training performance was modeled from cognitive abilities in Ref. [200].

Considering these principles, the effect and relationships between these components are elaborated. The model structure constructed in the stock-flow diagram in the Vensim environment can be found in S5 Table of the Section: Supporting information.

The personal profile consists of variables from "dynamic", "static", and "stress-related" profiles, and each of them is a look-up value from the corresponding curve regarding the assumed personal characteristics of our imaginary operator in the previous section. These functional curves are constructed from relevant literature. Besides interfering with either summation or canceling out the effect of each other, the variables here directly reduce the "initial personal capacity"

components linearly, so that these capacities will start from lower than their normative value. The same mechanism is chosen for the "circumstantial stressors".

Due to the assumed environmental effect, the variables within the "circumstantial stressors" will either increase or not affect the corresponding "task load" component, while also adding more value to the degradation rate of "personal capacity". For the sake of simplicity, only linear and additive effects of these factors are considered. Non-linear and multiplicative effects are not the focus of this use case; however, they can be constructed with the same model architecture by incorporating more elaborate functions.

Due to this construction, the perceived "workload" components are the numerical value of relative differences between corresponding pairs of incoming "task load" and "basic task load", and the "average situational demand" is simply the average value of all "workload" components. The "average capability degradation" is calculated similarly from the degradation of every work capacity. This pair comparison structure is inspired by the model of Karasek [201], and is also applied again in the evaluation of "perceived situational demand" and "perceived capability degradation", by comparing the calculated value with the corresponding personal "demand threshold". Consequently, an incoming task will have either a value of 1 as demanding if both "perceived situational demand" and "perceived capability degradation" are higher than personal norms, and 0 as no demanding on all other cases.

Then, the stress values of "sustained attention", "acute stress", and "chronic stress" are accumulated by one unit value for each simulation time unit, when the previous state is activated, and its value exceeds a certain threshold. For example, while the task is deemed as demanding, "sustained attention" is accumulated. When the "sustained attention" value reaches a threshold and keeps accumulating, then "acute stress" is activated. Each stress type affects the "workload" components, or capacity degradation rates by subtracting a ratio of stress value linearly from the current values. Due to this loop, the next incoming task has a higher chance of being perceived as demanding if the operator is under a positive stress effect, and vice versa. The components of "personal performance" are simply the addition of supporting factors, with the deduction of resisting factors, and the final OEE value is a multiplication of these components.

The look-up equation for these variables and their explanations from relevant studies, with the values/equations for immediate variables/stocks/flows, will be provided in more detail in S5 Table of the Section: Supporting information. Due to the nature of this conceptual model, a mixture of data from theoretical research, simulation, and experiments was used. When the equation of this system dynamic model is backed up with experimental data, relevant details such as study design, participant population, and data collection methods are also mentioned. Since no study mentioned the exact scope of the use case, only the general and unitless shapes of the curve or the function from these studies are incorporated into the model. When theoretical models and simulations with no actual data were utilized, only general trends and directions of effects were considered. For each specific case, the exact numerical values and thresholds may vary and can be subject to change. For the sake of simplicity, a non-denationalization technique is used, with partial removal of physical dimensions from its equations. Besides, the stress-related variables ("sustained attention", "acute stress", and "chronic stress") are not real values with physical meaning, but variables that represent the accumulation and relaxation behavior of these stress types.

A random function is used to generate the "task load" components, and the arrival time of new tasks is created randomly around a "basic work pace" of three minutes. The fixed duration of the simulation is 8 hours (480 minutes), with a 15-minute lunch break. Four scenarios were simulated:

- Working with a normal schedule: In this scenario, the "task load" components are generated randomly but close to the "basic task load". It is expected that the worker can maintain the performance during the work day with a fairly remaining "personal capacity".

- Working with high workload: This scenario introduces a "task load" that is higher than the "basic task load", or the worker has a physical impairment that leads to a shortage of working capacities.

- Working with additional breaks: The first intervention is to provide a ten-minute short break after every hour to prevent stress accumulation and facilitate relaxation.

- Working with a reduced work pace: The second intervention is to reduce the "task load" at the end of the working day, by reducing the work pace, to prevent the above-mentioned mild stress.

Fig 9 exhibited the task schedule from different simulated scenarios, where a value 1 means ongoing tasks and 0 means no task. The first and second scenarios used the same normal schedule, with a single lunch break in the middle of the working day. The third and fourth scenarios utilized interventions, such as additional breaks and reduced work pace, respectively.

## Simulated results

In the first scenario, the model successfully reflected the relationship between "task load", "workload", and "personal capacity", with different stress accumulation and relaxation behaviors. Fig 10 exhibits the simulated response from an

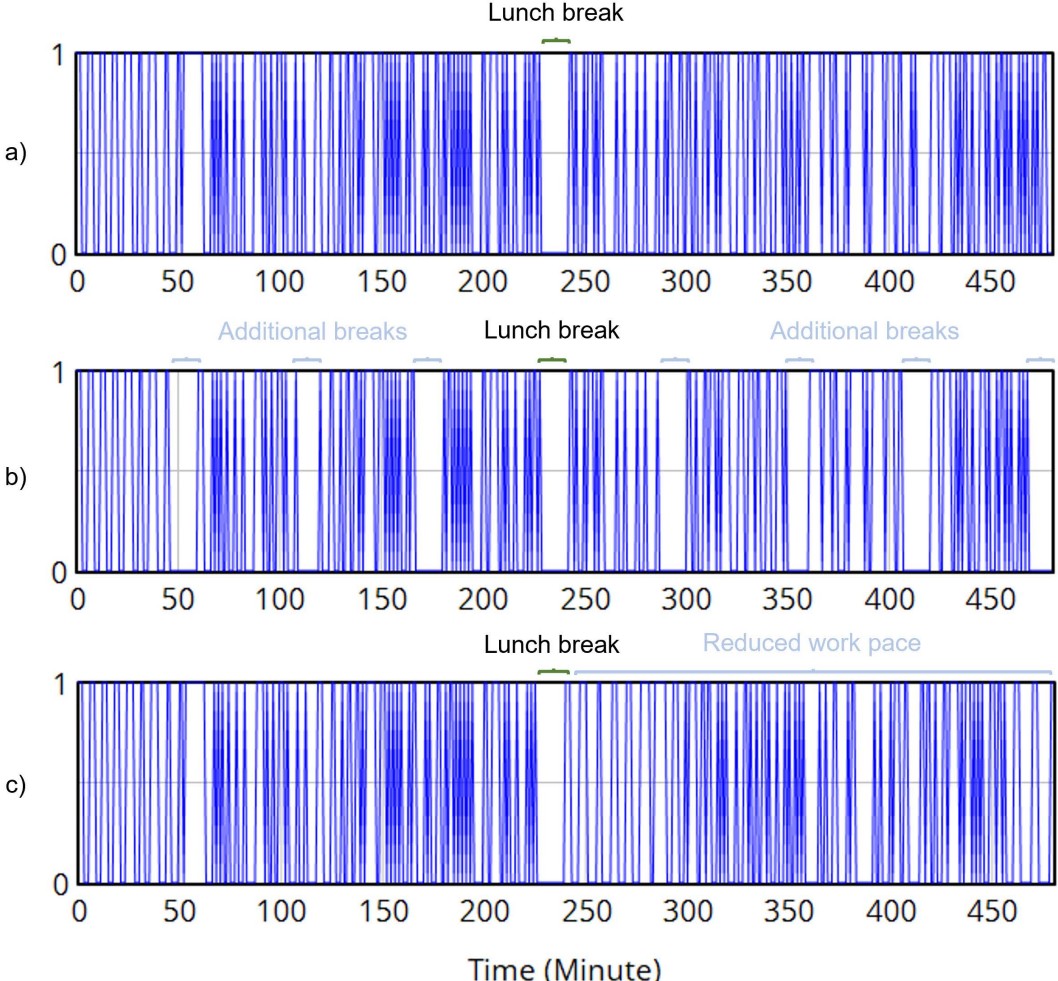

**Fig 9. The simulated working schedules in the use case, with two shifts with a lunch break (a), with additional hourly breaks (b), and with a reduced work pace after the lunch break (c).**

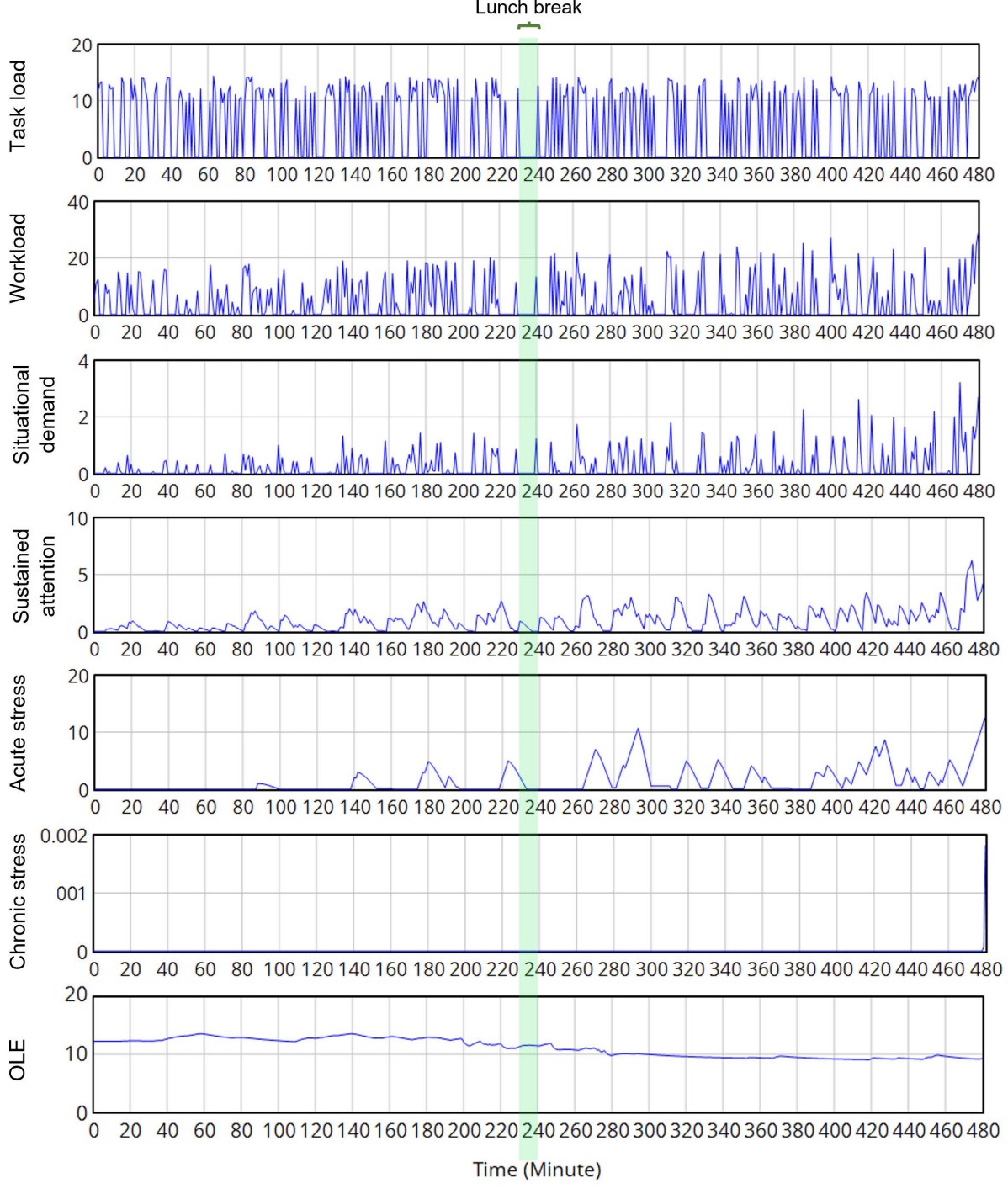

**Fig 10. The work behavior of the worker "A" in an 8-hour working day in the "normal load" scenario.**

arbitrary worker "A". Our worker started the work session with a full initial "personal capacity", with increasing attention and interest in the task. Due to exposure to demanding tasks, the "Sustained attention" level was accumulated slightly, especially during prolonged tasks due to problems or failures. "Acute stress" appeared several times when the "Sustained attention" was higher than the threshold. However, "A" has enough time between ongoing tasks and demanding situations to get relieved, thus an "optimal performance" peak can be recognized in the bell-shaped OLE curve. The OLE was stable at the beginning of the morning shift, though slightly decreased due to the work capacity degradation. Due to the positive effect of "Sustained attention", the OLE reached an optimal level between the 60th to 180th minutes before lunch break, when the work capacity is still sufficient, and the "sustained attention" level is within the personal threshold without any extra stress. Noticeably, the "Acute stress" and "Sustained attention" were still relaxing during the lunch break. After the lunch break, though the task loads were still the same, the performance kept decreasing, which led to a negative effect of "Acute stress". Due to the "natural degradation", "A" has utilized all of the work capacities, making the performance degrade slightly. At the end of the working day, "A" perceived an increasing workload with a high number of demanding tasks, a mild "Acute stress", and a very infinitesimal level of "Chronic stress". Without a long-term stressful feeling about the job, this mild stress status can be relieved with reasonable after-work activity and diet, with good sleep to regenerate working capacity and vigor [202]. If the working day lasts longer (i.e., overtime hours), even with the same amount of workload (not to mention the occurred failure/problem), then our worker will encounter the "Fatigue" status. In this case, the stress symptoms will exaggerate with significantly decreased performance [203], with a remarkable sign of "Acute stress", or even "Chronic stress".

In the second scenario, the "sustained attention" can not be accumulated most of the time, but "acute stress" was aroused and did not cease, thus the signs of "overload" and "fatigue" appeared as can be seen in Fig 11. "A" experienced demanding situations from the early time of the morning shift, with the "Sustained attention" built up quickly and reached a higher level than the personal threshold. "Acute stress" existed almost all of the morning shift, with the sign of "Chronic stress", suggesting an "Overload" status. This situation stops escalating during the lunch break, but gets worse during the afternoon shift. The OLE performance curve slightly increased in the morning when "A" encountered the first hour of hardship, but degraded much faster until the end of the day, which can be due to the extremely stressful perception when the worker feels the tasks are beyond current capability and control. As the worker kept working, the OLE kept decreasing noticeably, and a low value of "personal capacity" remained after a working day. The sign of "Fatigue" appears at the end of the working day, with a high level of chronic stress shown, whose effects last for a longer period and affect the working capacities, reducing the vigor for the next day, and making the worker more vulnerable to "burn-out". It can be concluded that the work is too stressful for the capabilities and experience of the worker. The more serious the negative impact of "fatigue", the harder the recovery from the on-job effort [204]. In reality, this scenario happens when a high task load is assigned to an inexperienced worker, or when a temporary physical impairment happens that reduces the initial working capacity, or when an uncomfortable physical environment condition occurs, a negative impact of "Overload" can happen immediately. Working for a long time in this condition ensures a "Burn-out" in occupational life, with a long-term reduction of "Personal capacity". This scenario emphasized the importance of a well-designed job with tailored work content for individuals.

The third and fourth scenarios showed the capability of the proposed model with stress-relieving interventions, which were designed as a work-content modification. The simulated results for these scenarios are exhibited in Fig 12. When additional breaks were added hourly, the number of tasks naturally reduced, which led to less demanding situations. The "Sustained attention" did not have enough time to accumulate, and the worker did not have enough attention and preparation for the coming tasks. Thus, the worker perceived a higher value of demand requirement and is more vulnerable to demanding tasks, which results in a higher peak value of "Acute stress", which harms the cognitive performance and further degrades the work capacity. The first intervention with additional breaks did not improve the performance of our worker but rather created the "under-load" status with a reduced "sustained attention" level; thus, more demanding

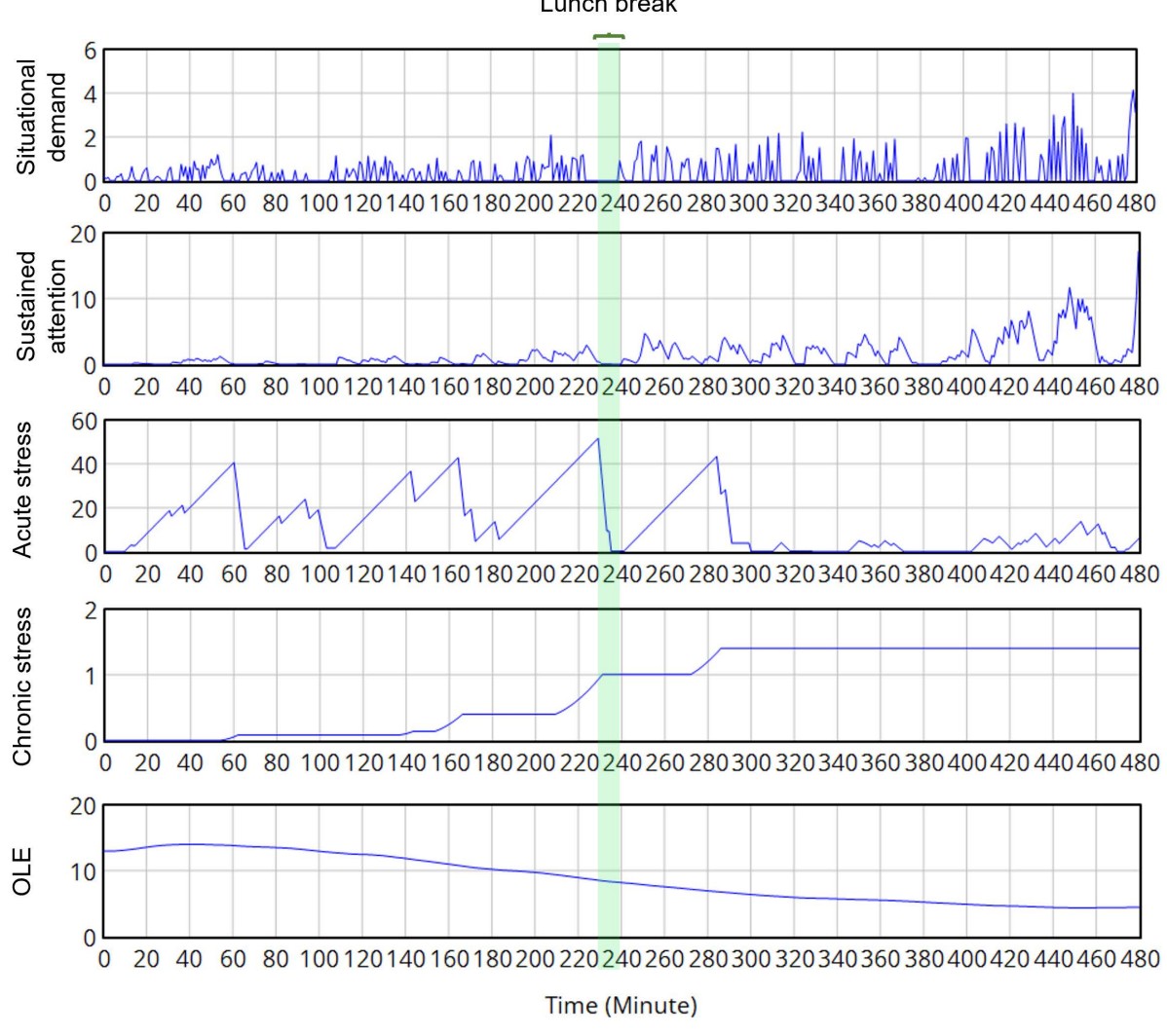

**Fig 11. The behavior of the worker "A" in the "overload" scenario throughout the 8-hour working day.**

situations can be perceived due to a lack of preparation and focus. The OLE of the worker of interest constantly decreases, due to the natural degradation of working capacities and lack of quality attention. Though this intervention created no "chronic stress", the performance was continuously degrading, which can harm the safety and long-term well-being [205].

In the fourth scenario, when a reduced work pace is applied after the lunch break, the worker has time to rest between incoming tasks in the afternoon shift. Though the tasks were still demanding and the working capacities were already decreased in the afternoon shift, the "Sustained attention" was still building up while having enough time between coming tasks to relax, and had enough time to accumulate and exert its positive effect. Only a trivial sign of "acute stress" appeared, and no "chronic stress" was left at the end of the working day. The OLE of our worker even slightly increased in the afternoon shift, which can be explained by a high vigilant attention level, with enough time for the muscles to rest and reflect upon coming tasks. In practice, this intervention can have a form of rotating into less demanding work positions.

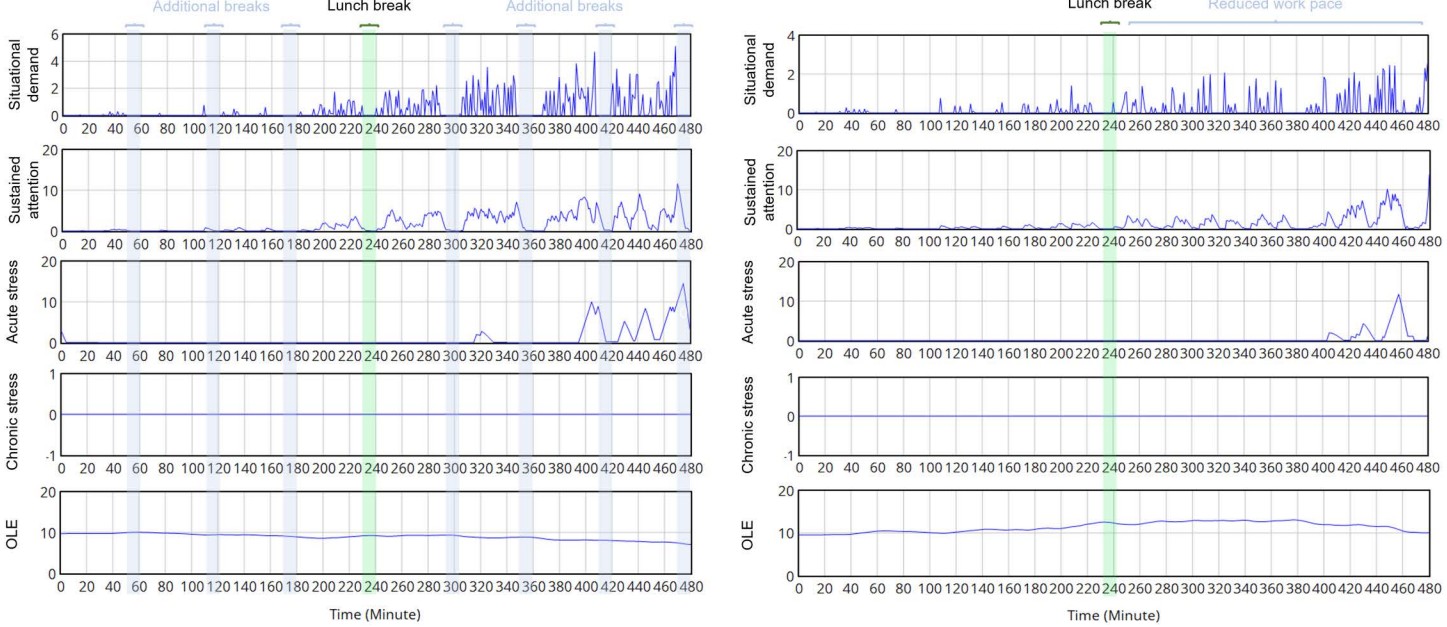

**Fig 12. The work behavior of the worker "A" in a working day with introduced interventions of additional breaks added hourly (left) and reduced work pace after the lunch break (right).**

Since parameters of system dynamic models are subject to uncertainty, sensitivity analysis is necessary to confirm the reliability of simulation results. Though Vensim supports this function in the licensed version, due to the limitation of the academic version in this use case, the constructed model is migrated into the Python environment with the PySD package [206]. Then, the sensitivity analysis is performed in a Markov chain Monte Carlo simulation with variables generated by the PyMC package [207]. Since the model has many parameters used for the configuration of the model, only some key input parameters are chosen to perform the sensitivity analysis for the "Sustained attention" in the use case. Fig 13 shows the sensitivity of the simulated "Sustained attention" to the variations in the age of the operator, and the time between failures. Each of these scenarios is repeated with a 200 simulation in the normal schedule, with the same workload components. It can be seen that age intermittently affects the perception of demanding tasks and thus increases the attention level during the interval of incoming tasks, as older operators tend to perform worse than younger colleagues due to their reduced attentional capacity [208]. On the other hand, the variable "failure rate" heavily affects the "Sustained attention" regardless of the incoming tasks, thus accumulating this value from the shift beginning until the end. This procedure can be reproduced similarly for other parameters. Besides serving the purpose of testing behaviors and robustness of the models, this analysis can be useful in adjusting the model architecture to be more compliant with real-life expectations.

## Discussion

Improving human performance while maintaining the balance between workload and induced stress is always a target in human resources management. Considering employing a worker with unique work capabilities and characteristics in a safe indoor working environment with a favorable work context, the reasonable amount and combination of work-content factors for this particular individual are an interesting research topic of all times [101,209]. With the advances of sensors, wearables, and big data analysis, the human stress status and produced performance are ready for real-time monitoring [22,210]. However, a computational model that can reflect the effect of the designed work content before the practical implementation will always be useful, as it provides a base expectation during simulation. Current models are either in

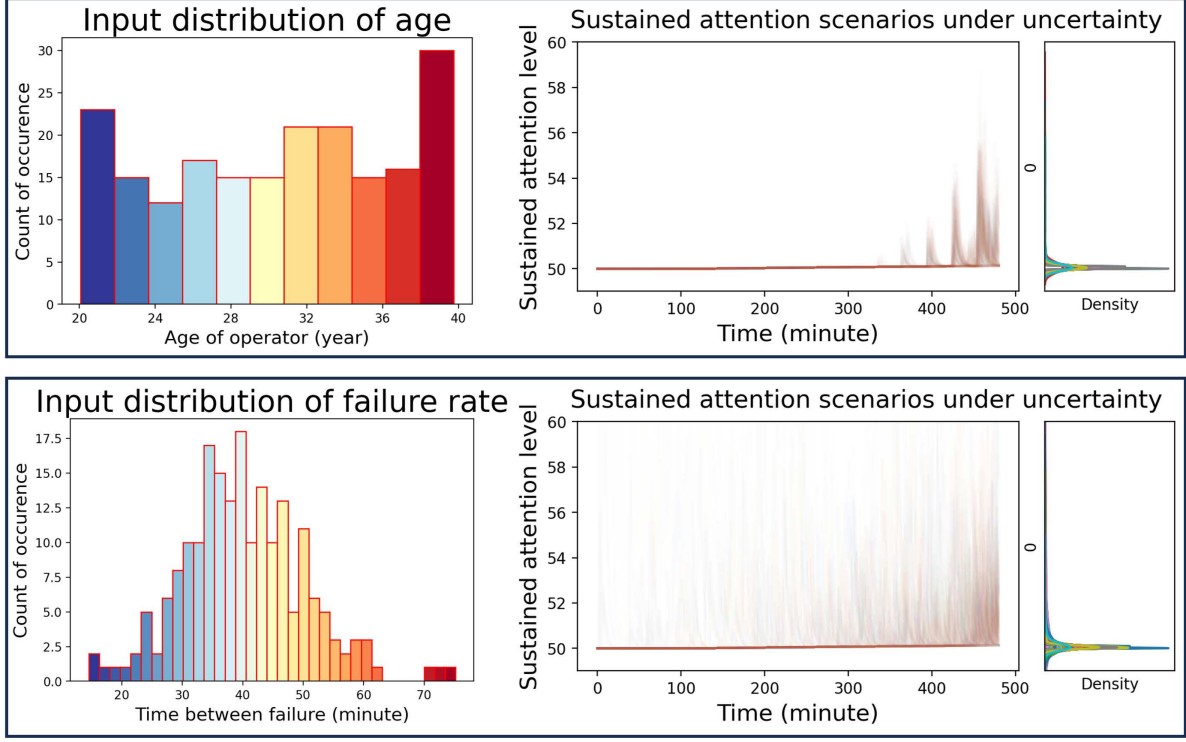

**Fig 13. The sensitivity of the "age" of operators on "Sustained attention" (upper), with the uniform distribution from 20 to 40 years old.** The sensitivity of the "failure rate" on "Sustained attention" (lower), with the normal distribution between the mean value of 40 minutes for each occurring fault.

lack of work content configuration, personal customization, or stress mechanisms for performance prediction, thus limiting further practical application.

The proposed conceptual model in this paper contains several useful functions due to its novel approach. Throughout four simulated scenarios, the results from the proposed model sufficiently meet the expectations from the literature. Not only are the subtle aspects of work behavior under stress reflected, but the effect of different interventions can also be illustrated. The achieved results can be used to understand the performance of each individual, therefore adjusting the work content. On the other hand, some limitations should be considered as available rooms for future development.

## Model usage

Though the model is constructed based on the diagnosed literature, its factors and scope are not fixed within these boundaries. The main concept is a flexible architecture that can be expanded and incorporate more factors. Though personal performance is mentioned, the model can be adjusted, and a group of workforce population with similar characteristics can be "digitalized" based on the same procedure. Besides proposed parameters and their directions of effect, additional modifications should be considered based on different application contexts, such as the characteristics of the workforce population or the nature of the work. Other aspects (e.g., the effect of a learning curve or skill decay) can be examined similarly.

From the workforce population aspect, the gender difference does not influence the perceived fatigue between male and female workers [211], therefore, the same acceptable physical load can be applied. However, the effect of workload on functioning ability between genders varied in the association of age [212]. Necessary adjustments with demographic/

occupational variables can be carried out and validated for a certain workforce population, e.g., Taiwanese and Western populations can sustain the same physical load and MAWT [131].

Regarding work nature, relevant parameters such as task load, cycle time, personal workload threshold, and the temporal and sequential relationship between tasks are indicated as required for validating the VACP model [93]. Different work shift duration requires a new setup to measure the "basic task load", e.g., the workload for a 4-hour shift can be 10% higher than for an 8-hour shift [131,213]. While the need to measure the "basic task load" is critical, personal customization is also important, as the same task load can impose different workloads on different individuals. A weak association between elements of psychological workload and overall body weight is suggested by Overgaaet et al. [214].

To validate the results from the model, general trends and behaviors predicted after simulation can be qualitatively compared with other empirical data from relevant studies, which also have a stable work environment and work context, and the effect of work content can be separated and isolated as the major stressor during work. Though it will be more challenging to test the probability of each OLE factor, the cognitive, emotional states, and stress levels are easier to validate with questionnaires and physiological parameters [21]. On the other hand, the model can also be validated with well-controlled experiments, in an appropriate work setup and suitable design of the experiment plan, with discrete and configurable work content factors. Since the description of designing such an experiment is too complex and out of the scope of this theoretical paper, our future research aims at creating such an experiment, similar to the case of the WEBA dataset [215], but preferably in the industrial environment.

Last but not least, to maintain a small prediction error during model usage, the model should be updated with recent input data from the personal profile. However, different factors have different update intervals, such as the age effect, which can be adjusted yearly, while job motivation is assessed quarterly, sleep quality should be considered weekly, and any physical impairment can be reported instantaneously.

## Model capability

The proposed approach, with a conceptual model, enables human-centric work-content-related planning, simulation, monitoring, and improvements. Therefore, the model is expected to support related work design, simulation, monitoring, and human-centric development. Well-designed "task load" can utilize the working capacities of workers, without compromising their physical and mental health.

Firstly, the model can capture the detailed relationship between each work-content factor and its effects on the physical and mental status of operators, industrial managers, and engineers, enabling them to diagnose the simulated results and identify the most suitable work-content combination that better utilizes an individual operator. Given a unique "personal profile", an optimal level of task load should have reasonable but not too high task demand, decent task variety or pattern change, and available social support to motivate the worker to voluntarily engage with the tasks [216,217].

Secondly, through the loop of planning and simulation, industrial engineers can design and arrange the work environment and the configuration of an allocated workstation to find the optimal setup. By controlling the relevant factors and adjusting the interested element, experiments can be designed to check the causality and effect of each work configuration on the perceived stress of the worker in a complex manufacturing system, such as the number of WIP in the product-mix production system [218].

Thirdly, the work-content conceptualization of the proposed model can pave the way for real-time work-content monitoring with timely intervention, by foreseeing the effect of possible work-content modification and comparing the expected performance under different work conditions. The developed mechanism that reflects stress-influenced behaviors can be used as a building element in the future Human Digital Twin (HDT) with human details and functional status [219,220], which further integrates humans into the H-CPS [221] and supports production supervisors in their daily monitoring tasks to harvest optimal human resource utilization.

For a long-term vision, after constructing the model with an individual "personal profile" and qualitatively fitting the simulated result with the observed actual performance, a human-centric development strategy can be elaborated. Besides short-term personalized work content and interventions, long-term initiatives such as personal training to improve skill acquisition and avoid skill decaying [222], strength training to prevent workability deterioration [223], skill development plan, customized assistance for impaired or disabled workers [224], and stress management, to broaden the stress endurance, enhance the positive stress effect and the resultant work efficiency for that individual. On the scale of workforce development, companies can have a benchmark of the optimal work content and configuration for the majority of their workforce, while workers have the opportunity to learn about their work capacities, strengths, and weaknesses, which helps them choose a suitable work schedule and lifelong career for their work-life balance.

### Model limitation

There are limitations associated with the natural characteristics of building elements and assumptions in this conceptual model. The first and foremost limitation of the proposed model is the simplification of many interactions and relationships, e.g., the individual and interacted effect of stress from "task load" to "workload", the interaction between "workload" components, the perception of stress from "workload", and the effect of stress on work performance. Many relationships are non-linear and multiplicative, but are simplified as linear and additive in the use case. However, developing a comprehensive model that is capable of reflecting all subtle aspects of human psychology is impractical within the scope of this study. The authors proposed this model structure and mechanisms with a certain level of simplification, and also called for more innovative interdisciplinary approaches to diagnose these interactions in a complex system of human perception and workloads.

The second limitation is the lack of reference data with quantitative measures and numerical thresholds during the model development. This fact not only obstructs the validation of the model in real life but also limits it to a qualitative tool, and poses a hurdle to its application context. To the knowledge of the authors, there is no experiment or any real-life use case that have similar construct (task load components, induced OLE from work capability, etc.) and can be used for quantitative validation. Among the reference data and experiments that were referred to in the Section: Supporting information, many experiments are conducted with different research purposes and incomparable conditions, thus hindering the generalization of these studies for other conclusions. Though there are different scales and measurements for workload assessment, physically and mentally [225], however, the quantified association between these scales is not available, nor is there a relative comparison between them. Many factors have well-known directions of effect (i.e., years of experience, age), but their representative curves lack quantitative milestones, for instance, the number of products in the learning curve. The inhomogeneous measurement of factors (i.e., some are measured by physiological parameters or bio-markers, some by questionnaires) is challenging in both model development and usage phases. Several variables do not have specific measurement scales, such as stress endurance and personal stress threshold. Though these concepts were used intuitively in many studies as personal stress characteristics [226,227], the authors did not find any official definition nor validated measurement for them. The accumulation/relaxation rate of different types of stress has also not been elaborated in the previous literature. Consequently, the model is not able to predict the exact stress level in numerical results, according to current popular stress measurement scales. Our next work will be validating the structure of this conceptual model with expert opinions through interviews and targeted surveys. However, users can qualitatively predict the increasing or decreasing behavior of stress and performance status of each individual, as a time-based function. On the other hand, as this study focused merely on the conceptualization of the proposed model, evaluation metrics are not elaborated at this stage. To overcome this limitation in future research, user studies on the relationship of perceived workload and impact on task performance (e.g., [228]) will be deployed, while relevant objective and subjective measurements for each type of workload (provided in S2 Table in the Section: Supporting information) can be used to evaluate the simulated results from the model.

As a personal profile plays an important role in setting up this model, the third limitation is the significant personal customization for each individual regarding their unique physiological features. It is widely accepted that years of experience have a positive effect on performance; however, the scale of this effect on individuals is unclear. Not to mention the basic workload should be measured based on the natural ability of each worker, the sensitivity and relaxation rate for each type of stress is also dependent on the cortisol level regulation of each individual [229], which is the close influence of the HPA axis [230]. For special tasks that require sustained attention, characteristics such as state and trait self-control may have more influence, though the effect has not been concluded [231]. The efficiency of stress recovery is also affected by working hours [232] and environmental noise [233], which are varied individually, and reflect different symptoms of stress burden on a single person [234]. Low self-esteem, stressful life changes, and recent minor life events can also affect individual stress recovery under acute stressors [235]. These associations require in-depth, detailed personal customization regarding historical medical and stress disorder records; therefore, they are not included in this study.

Another limitation is the limited amount of high-level evidence (e.g., Randomized controlled trials, well-designed experiments, or datasets), which leads to uncertainty in the effect direction of each work content factor and theoretical assumption. There is a lack of in-depth analysis in the interaction between physical and mental workload on different age groups of types of mental and physical activities [89,236], and their corporate effect on human perceived workload and performance. The stress effect of work content in experiments conducted in different work environments and contexts (e.g., traffic control [237] versus industrial quality inspection [238]) is not comparable, thus requires different follow-up studies for later confirmation with the simulated results in each case. To validate the human status, a combination of both objective and subjective measurements is suggested [123], as the use of a sole physiological parameter such as HRV to validate the AWCRS is still in the immature phase [45], which is not fully suitable for practical implementation. The effectiveness and impact duration of proposed interventions should also be validated before implementation during the simulation, as they yield different effects under various usage contexts, such as caffeine and caffeinated energy drinks can have both positive (e.g., increased alertness, reaction time, and cognitive performance [239]) and negative (e.g., decreased sleep quality [240,241]) effects with unknown risk/benefit ratio [240]. The job rotation intervention can pose different effects within different work settings [242]. The generalization of current relevant studies that studied the effect of intervention should be analyzed based on their experimental design.

In a practical manufacturing environment, not only the personal performance be considered, but the collective performance of workers in a manufacturing line should be investigated. Different workers with different stress effects may affect collaborative work [169]. However, in the scope of this research, individual performance was focused, with the assumption of collaboration as the support readiness level that the worker receives from co-workers and supervisors. The collective performance requires more model setup and constraints; thus will be aimed at in our future research.

Last but not least, since this study focuses on building a theoretical concept, its usage has more meaning in qualitatively explaining and generalizing the human work behaviors in the industrial environment, rather than generating empirical results that can be readily replicated. However, general trends and behaviors predicted from the model simulation can be qualitatively compared with other empirical data, if these studies fulfilled similar assumptions of the model, such as work environment and work context are stable and have a static effect, and the effect of discrete repetitive work content can be separated and isolated as the major stressor during work. Though it will be more challenging to test the probability of each OLE factor, the simulated stress levels are easier to validate with questionnaires and physiological parameters that are mentioned in the Supplementary Materials. To have data that best suits the purpose of empirical validation, the authors also suggest the construction of experiments in a well-controlled environment, with an appropriate work setup and a suitable design of the experiment plan, so that the effect of work content factors can be configured and captured. Since the description of such an experiment is too complex and may be out of the scope of this theoretical paper, the authors plan to publish the design guidelines and examples in separate studies.

## Conclusion

Integrating human factors into a Human-Cyber-Physical System (H-CPS) was crucial in I5.0. This research proposed a conceptual model to reflect the acute stress of industrial workers under the effect of work-content factors and predict their OLE performance. Besides structuring the complex relationship between work-content factors and induced stress, the authors also try to conceptualize personal performance in the industrial work environment. With an interdisciplinary perspective, the incorporated effects reflect many subtle aspects of worker behavior when receiving work-content elements as stressors. Despite several intrinsic limitations due to the lack of specific quantitative units and scales, the model helps to understand the physiological reaction of an individual qualitatively under certain work conditions, thus enabling better production planning and preparing stress management initiatives, to optimize the human resources utilization.

A qualitative model is built in the Vensim environment to validate the proposed concept. Four simulated scenarios with two work-content-related interventions are experimented with in this use case. The result shows that the model can reflect the subtle status and behavior of different stress responses and performance levels aligned with common knowledge and suggested from previous literature, under the incoming work content, within a specific work setting. Though the simulation is limited to four scenarios, the positive results show the usability for work-content planning, simulation, monitoring, designing interventions, and preparing human-centric developments. The usage of this model results in a better understanding of human worker capacities while attracting more attention from industrial managers to create a favorable work environment, regarding the interaction of their workers with the working conditions, line setup, and task requirements. In further work, the authors will apply this model in more complicated scenarios to assess its capability to align with known phenomena.

Besides the discussed limitations of the developed model, this study has some limitations regarding its scope. Not only was the stress-inducing mechanism simplified, but more subtle interactions from factors such as the training effect or learning curve were also ignored. In addition, not only the individual performance but the collaboration between individuals and social interaction with others should be elaborated, given the fact that no operator works individually in an isolated environment. Some of the future works will aim to alleviate these, for example, investigating the effect of learning on the perception of workload, given the fact that the cognitive workload during training may differ based on task load [243] or personal profile of the trainee [244]. The perception of workload will be constructed with different mechanisms for the collaboration between human [245], and human-robot [246] separately, as these intricate relationships require knowledge from more disciplinaries. Besides, the authors urge for more well-constructed experiments to validate the hypothesized assumption about the effect of work-content factors on workload perception. A real-life experiment was conducted to collect actual scenario data of human performance under the effect of the work-content factors [215]; however, the individual and corporate effects of workload components are still overlapped and require separate experiments to validate.

As human-centric models that reflect stress-influential behaviors can enrich the future HDT development with human details and functional status [219,220], further detailed development is expected to integrate humans into the H-CPS [221]. Though the concept of a personal HDT is proposed in the healthcare industry and fitness management [247,248], the manufacturing industry still lacks attention and motivation for similar personalization and detailed simulation mechanisms, probably due to more focus on economic growth [87,249]. Despite many simplifications, this study attempts to further endorse human centricity in the I5.0, by connecting the dots from different fields to quantitatively assess the dynamic effect of work-content factors on the performance of industrial workers. The authors also proposed several future research directions to improve the current work so that it can be mass-customized for the workforce, serving not only the personnel but also workforce development, thus yielding a large-scale influence. These research efforts will bring a good scientific foundation for human resources management and utilization in the ongoing Industry 5.0.

## Supporting information

**S1 Table. The glossary of relevant terms [250–327].** Terms that are used in the proposed model are explained in the Table below. The terms are sorted in the order of appearance throughout the work.
(PDF)

**S2 Table. The proposed scopes of physical and mental "task load" in the proposed model.**
(PDF)

**S3 Table. Different factors considered in a personal profile during the model development.**
(PDF)

**S4 Table. Circumstantial stressors and their effects from the literature.**
(PDF)

**S1 Fig. The initial personal capacities, with the static effect of the "Personal profile" on the initial "Personal capacity" of a worker at the beginning of the working day.** For better comprehension, the model structure in the Vensim environment was developed in separate views, with each view represented in the following figures.
(TIF)

**S2 Fig. The dynamic effects of circumstantial stressors.** The dynamic effects of circumstantial stressors on the perceived workload and working capacity degradation.
(TIF)

**S3 Fig. The task perception with different task load components.** This Fig visualizes the model structure to generate task load components, and how the work capacities are modeled by stock variables.
(TIF)

**S4 Fig. The stress accumulating mechanism during a working session.** This Fig describes the structure to create the accumulation and relaxation mechanism of different stress types and their effects on workload and capacity degradation.
(TIF)

**S5 Fig. Personal initial work capacities are defined based on the personal profiles.** The OLE is defined by its constituents.
(TIF)

**S5 Table. Model variables, values, and equations.** The details of the use case setup with the important variables, along with their values and equations. Relevant experiment data or theoretical models used are mentioned. Given that no study mentioned the exact scope of the use case, only the general shapes of the curve or the function from these studies are incorporated to generate the inputs of the model. Other variables were constructed with simplified relationships from input variables, based on reasoning from relevant theoretical studies.
(PDF)

## Acknowledgments

The authors express their gratitude to Dr. Andrea De Gaetano and Dr. Antonio Cerasa, scientists from the IRIB-CNR Institute, Italy, for their helpful comments.

## Author contributions

**Conceptualization:** Tuan-anh Tran, Tamás Ruppert, János Abonyi.

**Methodology:** Tuan-anh Tran, János Abonyi.

**Software:** Tuan-anh Tran.

**Supervision:** György Eigner, János Abonyi.

**Validation:** Tuan-anh Tran, Tamás Ruppert, János Abonyi.

**Visualization:** Tuan-anh Tran.

**Writing – original draft:** Tuan-anh Tran, Tamás Ruppert.

**Writing – review & editing:** György Eigner, János Abonyi.

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
