## [Decision Letter · Decision Letter 0]

12 Dec 2025

PONE-D-25-57202Conceptual Qualitative System Dynamics Model for Simulation of Perceived Workload, Stress and Performance from Industrial Work ContentPLOS One

Dear Dr.Ruppert

Thank you for submitting your manuscript to PLOS ONE. After careful consideration, we feel that it has merit but does not fully meet PLOS ONE’s publication criteria as it currently stands. Therefore, we invite you to submit a revised version of the manuscript that addresses the points raised during the review Please ensure that you address all of the reviewers’ comments below.

If applicable, we recommend that you deposit your laboratory protocols in protocols.io to enhance the reproducibility of your results. Protocols.io assigns your protocol its own identifier (DOI) so that it can be cited independently in the future. For instructions see: https://journals.plos.org/plosone/s/submission-guidelines#loc-laboratory-protocols. Additionally, PLOS ONE offers an option for publishing peer-reviewed Lab Protocol articles, which describe protocols hosted on protocols.io. Read more information on sharing protocols at . Additionally, PLOS ONE offers an option for publishing peer-reviewed Lab Protocol articles, which describe protocols hosted on protocols.io. Read more information on sharing protocols at https://plos.org/protocols?utm_medium=editorial-email&utm_source=authorletters&utm_campaign=protocols..

We look forward to receiving your revised manuscript.

Kind regards,

Gerard Hutchinson, MD

Academic Editor

PLOS One

Journal Requirements:

“”Nemzeti Kutatás, fejlesztés és innovációs hivatal (NKFIH)

“NO authors have competing interests”

5. We note that your Data Availability Statement is currently as follows: “Not applicable, no related data”

Please confirm at this time whether or not your submission contains all raw data required to replicate the results of your study. Authors must share the “minimal data set” for their submission. PLOS defines the minimal data set to consist of the data required to replicate all study findings reported in the article, as well as related metadata and methods (https://journals.plos.org/plosone/s/data-availability#loc-minimal-data-set-definition).).

Reviewers' comments:

Reviewer's Responses to Questions

**Comments to the Author**

1. Is the manuscript technically sound, and do the data support the conclusions?

Reviewer #1: No

Reviewer #2: Yes

2. Has the statistical analysis been performed appropriately and rigorously? 

Reviewer #1: I Don't Know

Reviewer #2: No

3. Have the authors made all data underlying the findings in their manuscript fully available?

Reviewer #1: Yes

Reviewer #2: No

4. Is the manuscript presented in an intelligible fashion and written in standard English?

Reviewer #1: Yes

Reviewer #2: Yes

5. Review Comments to the Author

Reviewer #1: The manuscript proposes a qualitative system-dynamics model aimed at simulating how industrial work content influences perceived workload, stress dynamics (sustained attention, acute stress, chronic stress), and performance. The topic is timely and highly relevant in the context of Industry 5.0, human-centric manufacturing, and Human Digital Twins.

However, the manuscript also has several limitations that significantly hinder clarity and scientific rigor. The model remains purely conceptual, with limited validation, unclear mathematical formalization, and a somewhat overwhelming level of complexity that may not be fully justified by its qualitative nature.

Major issues to be addressed:

1. Although the manuscript repeatedly uses terms such as “accumulation,” “flow,” “threshold,” and “dynamic effect,” the actual functional equations, update rules, or explicit definitions of stock–flow relationships are not presented.

2. The state-of-the-art related to the mental states evaluation in industrial environments should be updated (e.g., include and discuss: https://www.mdpi.com/2076-3417/15/4/1822)

3. Although the model is conceptual, several key constructs, such as “sustained attention accumulation,” “acute stress thresholds,” “capacity degradation proportional to stress,” or “natural relaxation rate”, are physiological processes with well-documented non-linear behaviors. The model presents them as linear, additive, or rule-based, but without empirical justification.

4. The authors present a use case implemented in Vensim, but no quantitative comparison with empirical data is provided. The claim that the simulated outcomes “returned similar phenomena proposed by relevant studies” is anecdotal and not supported by systematic evidence.

5. The model meticulously incorporates dozens of elements (seven task load components, seven personal capacities, dynamic and static circumstantial stressors, multiple thresholds, two types of time load, multiple stress transitions). Yet the simulation example simplifies many of these.

6. The use case describes an imaginary worker and workstation in detail but does not provide: formal model inputs, numerical trajectories of stress stocks, performance outputs, visualizations of key model behavior.

Minor issues to be addressed:

- Figures 1–8 are informative but dense. Consider redesigning for improved readability

- The review is very extensive, but some claims would benefit from more recent neuroergonomics and psychophysiology literature.

- Some terms are excessively overlapping (e.g., “perceived workload,” “interacted load,” “workload component interaction”). A glossary may help.

Reviewer #2: The paper addresses a highly relevant and appealing research topic, and the proposed approach is clearly described. However, to enhance the quality of the work, the following issues should be addressed before the article can be considered for publication:

• The resolution of all figures is very low; most of the text within them is unreadable.

• It is unclear whether the text below each figure is intended to be part of the caption. If it is part of the caption, it is overly detailed and not sufficiently focused on the figure itself. If it is not part of the caption, then its purpose and connection to the main body of the manuscript (particularly on page 11) are not explained.

• The section on page 14 begins with the sentence “To validate the proposed concept,” but it is not clear which concept the authors are referring to. In the preceding section, only the problem formulation is presented; no model or conceptual framework is introduced.

• The task function introduced on page 17 is based on a 480-minute simulation. This seems inconsistent with the assumption of a 30-hour weekly workload stated on page 15.

• The process used to generate the results is not explained. No numerical values are provided, and it is unclear what type of simulation was performed. The authors should clarify what is meant by “simulation” and describe the methodology in detail.

• The discussion section is very limited, and it is unclear which specific results form the basis of the authors’ interpretations.

• A synthesis of the major gaps filled by the manuscript, along with further development of the approach, could help focus the results of the scientific investigation and propose future directions for development.

In summary, the manuscript presents the following strengths and weaknesses: the first part of the manuscript is well written and provides a clear and comprehensive literature review on stress, performance, and task load. This section effectively summarizes the existing body of research and offers a solid foundation for the study. In contrast, the second part of the manuscript, the model description and simulation appear less clear and lacks essential information. It is not evident what type of simulations were conducted, nor are the parameter values or assumptions used in the analysis adequately described. The discussion is limited, relying heavily on “simulated results,” whereas concrete, empirically grounded results would be expected. Additionally, the final section on model implementation feels somewhat forced and does not present a convincing or practical application of the proposed model. I encourage the authors to substantially revise and strengthen the second part of the manuscript to enhance clarity, rigor, and overall scientific contribution.

6. PLOS authors have the option to publish the peer review history of their article (what does this mean?). If published, this will include your full peer review and any attached files.). If published, this will include your full peer review and any attached files.

.

Reviewer #1: **Yes:** Vincenzo RoncaVincenzo Ronca

Reviewer #2: No

---

## [Author Response · Author response to Decision Letter 1]

10 Mar 2026

Reply to Editor - [PONE-D-25-57202] - [EMID:ee5238a98b94b99f] (PLOS ONE)

Title: Conceptual Qualitative System Dynamics Model for Simulation of Perceived Workload, Stress and Performance from Industrial Work Content

Dear Editor,

We are glad to receive the positive feedback and would like to express our appreciation for your comments and suggestions. We have addressed the reviewer’s comments in the letter below. We hope that the modifications (marked with blue color in the manuscript) have significantly improved the quality of the manuscript.

Sincerely yours,

Author

Dr. Tamás Ruppert

Reply to Reviewer 1 - [PONE-D-25-57202] - [EMID:ee5238a98b94b99f] (PLOS ONE)

Title: Conceptual Qualitative System Dynamics Model for Simulation of Perceived Workload, Stress and Performance from Industrial Work Content

Dear Reviewer,

We are grateful for your constructive critique and valuable comments. We have addressed each of your comments below. Furthermore, we have asked a professional, native speaker to proofread the manuscript. We hope that the modifications (marked with blue color in the manuscript) have significantly improved the quality of the manuscript.

Sincerely yours,

Authors

The manuscript proposes a qualitative system-dynamics model aimed at simulating how industrial work content influences perceived workload, stress dynamics (sustained attention, acute stress, chronic stress), and performance. The topic is timely and highly relevant in the context of Industry 5.0, human-centric manufacturing, and Human Digital Twins.

However, the manuscript also has several limitations that significantly hinder clarity and scientific rigor. The model remains purely conceptual, with limited validation, unclear mathematical formalization, and a somewhat overwhelming level of complexity that may not be fully justified by its qualitative nature.

Thank you for pointing out the strengths and drawbacks of this study. As you indicated, and as we stated in the title, this model addresses the multifaceted characteristics of the stress-performance relationship in industrial environments. Due to its complexity, we would keep it as a conceptual model, which can serve as crude guidance for workload design and estimation. Despite its qualitative nature, we want to emphasize its possible theoretical contribution in practice. We broke down the details of your constructive comments and answered them as follows:

Major issues to be addressed:

1. Although the manuscript repeatedly uses terms such as “accumulation,” “flow,” “threshold,” and “dynamic effect,” the actual functional equations, update rules, or explicit definitions of stock–flow relationships are not presented.

Thank you for this constructive comment. To enhance the clarity of these constructions in the model, we re-emphasized how these concepts of “accumulation”, “personal threshold” are incorporated in the choice of model elements in the subsection “Model structure and simulated scenarios”. The “flow” and “dynamic effect” are defined at the end of the same subsection, as well as explained better in the next subsection of “Simulated results”, to show how the mechanism from theoretical concepts are reflected.

Furthermore, in Table “S2 Table. Different factors considered in a personal profile” and “S4. Model variables, values and equation” in the section “Supporting information”, additional explanations are added to emphasize how the dynamic effect of “thresholds”, “flow” and “accumulation” are incorporated in each variable/value, detailing which factor creating which effect. Regarding the equations, we also re-emphasized in the “Model structure and simulated scenarios” that the ones used in this use case are suggestions, based on the trends or general tendency from relevant experiment data, or theoretical models, since no reference study mentioned the exact scope of their study with quantized variables. Therefore, only the general and unitless shapes of the curve or the function from these studies are incorporated, while the exact numerical values and thresholds may vary and can be subjected to change. We hope this amendment improves the cohesion and readability of the paper.

2. The state-of-the-art related to the mental states evaluation in industrial environments should be updated (e.g., include and discuss: https://www.mdpi.com/2076-3417/15/4/1822)

Thanks for your suggestion, we have incorporated the findings of the mentioned study in the “Introduction” section. This amendment helps to improve the comprehensiveness of our paper, and provides a practical perspective on how the relevant human-factors can be assessed with quantized, subjective metrics. Furthermore, we recommended it once again in the “Discussion” section that the trends, or tendency from simulated results can be validated using the techniques mentioned in the reference.

3. Although the model is conceptual, several key constructs, such as “sustained attention accumulation,” “acute stress thresholds,” “capacity degradation proportional to stress,” or “natural relaxation rate”, are physiological processes with well-documented non-linear behaviors. The model presents them as linear, additive, or rule-based, but without empirical justification.

We highlight this matter in the “Problem formulation and Preliminaries” and “Model structure and simulated scenarios” sections, emphasizing that these behaviors can be set to be linear or non-linear, additive or multiplicative, depending on the intrinsic characteristics of the industrial tasks of interest and the context of the simulation scenario. However, in the use case, for simplicity, we only test linear and additive effects with thresholds of accumulation, drawing on references that we have incorporated. In the subsection “Model limitation”, we also mentioned that, although the use case is conducted in a simplified linear scenario, the model structure is well-designed for other nonlinearities and can be tested with a different set of input equations.

4. The authors present a use case implemented in Vensim, but no quantitative comparison with empirical data is provided. The claim that the simulated outcomes “returned similar phenomena proposed by relevant studies” is anecdotal and not supported by systematic evidence.

Thank you for this constructive comment that points out an ambiguity within our presentation. We have edited this sentence in the abstract. Since the purpose of the study is to build a conceptual model with structures that can exhibit the general pattern of trends, phases, periods, etc. of human statuses related to Overall Labor Effectiveness that were collected during the literature review, rather than point prediction of specific value, the simulated results cannot be compared to any specific numerical results quantitatively.

On the other hand, no study have the similar scope of task load or comparable construct of predicting “availability”, “productivity”, and “quality” performance. To validate the model’s simulated results, one must organize an experiment with a comparable conceptualization.

We admit this fact, and we have highlighted this matter before explaining the model structure in the “Model structure and simulated scenarios” subsection, and at the beginning of the “Simulated results” subsection. Since it is a conceptual model, we can only state that the chosen structure can incorporate all the necessary concepts and mechanisms, without quantitatively validating it. We stated in the “Discussion” section that our next work will be qualitatively validating the model’s structure with expert interviews and surveys.

5. The model meticulously incorporates dozens of elements (seven task load components, seven personal capacities, dynamic and static circumstantial stressors, multiple thresholds, two types of time load, multiple stress transitions). Yet the simulation example simplifies many of these.

The proposed architecture can be further elaborated with these elements. However, in this use case, the intention was to show that the model can be customized according to a certain problem. Therefore, we simplified some mechanisms with preliminary assumptions to focus on a minimum viable simulation. However, thanks for your notice, we also state in the conclusion that the model can become more complicated, deploying extensions on more factors, and utilizing all the mentioned mechanisms.

6. The use case describes an imaginary worker and workstation in detail but does not provide: formal model inputs, numerical trajectories of stress stocks, performance outputs, or visualizations of key model behavior.

We are glad that this comment helps us to further improve the presentation of the simulated results. According to your comment, more details have been complemented in the paper. We explained how we chose to translate the profile of the imaginary worker into model input in the “Description of industrial assembly line environment” subsection, and set the model initial parameters in the “Model structure and simulated scenarios” subsection. In “Simulated results” subsection, we explained the general trends, tendency of the stock variables such as “situational demand”, “sustained attention”, “acute” and “chronic stress”, and “OLE”. We hope this clarification contributes to a better delivery of the results.

Minor issues to be addressed:

7. Figures 1–8 are informative but dense. Consider redesigning for improved readability

These figures are simplified to contain only the key concepts without conveying too much background information. We hope this amendment improves the readability of the paper.

8. The review is very extensive, but some claims would benefit from more recent neuroergonomics and psychophysiology literature.

Thanks for this constructive comment, we have looked for and added some references from recent literature in these two fields in the “Introduction” section. We hope these newly added references help to establish an up-to-date theoretical foundation for the paper.

9. Some terms are excessively overlapping (e.g., “perceived workload,” “interacted load,” “workload component interaction”). A glossary may help.

Thanks for your comment, we have added a glossary of frequently used terms at the beginning of the “Supporting information” section.

Reply to Reviewer 2 - [PONE-D-25-57202] - [EMID:ee5238a98b94b99f] (PLOS ONE)

Title: Conceptual Qualitative System Dynamics Model for Simulation of Perceived Workload, Stress and Performance from Industrial Work Content

Dear Reviewer,

We are grateful for your constructive critique and valuable comments. We have addressed each of your comments below. Furthermore, we have asked a professional, native speaker to proofread the manuscript. We hope that the modifications (marked with blue color in the manuscript) have significantly improved the quality of the manuscript.

Sincerely yours,

Authors

The paper addresses a highly relevant and appealing research topic, and the proposed approach is clearly described. However, to enhance the quality of the work, the following issues should be addressed before the article can be considered for publication:

1. The resolution of all figures is very low; most of the text within them is unreadable.

We have updated all figures to enhance readability. For the first eight figures, some simplifications have been made to reduce noncritical background details. The texts in all figures are now modified with sharper resolution.

2. It is unclear whether the text below each figure is intended to be part of the caption. If it is part of the caption, it is overly detailed and not sufficiently focused on the figure itself. If it is not part of the caption, then its purpose and connection to the main body of the manuscript (particularly on page 11) are not explained.

Thanks for this comment, we have refined the caption of each figure, to make sure that the captions are sufficient to explain the details in the corresponding figures. Excessive information is moved to the main text of the paper, thus keeping the captions succinct.

3. The section on page 14 begins with the sentence “To validate the proposed concept,” but it is not clear which concept the authors are referring to. In the preceding section, only the problem formulation is presented; no model or conceptual framework is introduced.

Thanks for pointing out this ambiguity in the mentioned sentence. Indeed, it confuses as the readers may not know the true intention of the following part, and how it will be connected to the previous theoretical constructs. We have modified this sentence to suggest that the section “Human-centric stress-performance simulation” is intended to show “an example model utilizing the conceptual formulation and constructed mechanism mentioned in the previous section”. We also clarified that in the following subsection, step-by-step problem formulation and development of a minimum viable model are shown, with the structures and equations of that model given in the “Supporting information” section. Thanks for your comment, we emphasized that no new conceptual framework will be produced; only an example model will be developed for a simplified scenario. Despite the simplification in the use case, we believe that the theoretical contribution still lies in the comprehensive and systematic approach of collecting the necessary stress-performance mechanisms from the relevant research. The section “Model limitation” also reminds the readers that extension and modification can be made based on the same theoretical conceptualization, for example, non-linearity can be used instead of the current linear equation, to cover more complex physiological responses of workability reduction under work stress.

4. The task function introduced on page 17 is based on a 480-minute simulation. This seems inconsistent with the assumption of a 30-hour weekly workload stated on page 15.

Thanks for this notice, we have clarified the use case formulation, as this “weekly working hour” variable should be defined as the “accumulated working hours” of the operator in simulation in the week of interest. This variable is relevant to the starting working capability of the operator, and should not be confused with the 480-minute duration of a simulation session. The simulation can be conducted at different shifts or days in the week, in which the accumulated working hours will have different values, counting from the beginning of the week. We hope that this clarification improves the readability of the paper.

5. The process used to generate the results is not explained. No numerical values are provided, and it is unclear what type of simulation was performed. The authors should clarify what is meant by “simulation” and describe the methodology in detail.

We found your concern reasonable, and thus we have extended the section “Human-centric stress-performance simulation” with more clarification of the “simulation method”. Especially, more information regarding use case formulation is added in the “Model structure and simulated scenarios” subsection, while the subsection “Simulated results” has been enriched with more explanation on the results, how it is achieved, and compared qualitatively with other phenomena reported in other studies.

6. The discussion section is very limited, and it is unclear which specific results form the basis of the authors’ interpretations.

The “Discussion” section is extended with more explanation, which stemmed from the qualitative comparison of the simulated results, and the expectation from other relevant studies, thus serving as a foundation for our interpretation and arguments about the developed work. In addition, we have emphasized that our work is conceptual, with a model structure constructed based on concepts and mechanisms collected from relevant studies, and so far there is no experiment or any real-life use case that have similar construct (task load components, induced OLE from work capability, etc. ) and can be used for quantitative validation. This limit is naturally due to the fact that human response and performance under stress is a complex problem, which requires more development and construction on any simulation mode

---

## [Decision Letter · Decision Letter 1]

26 Mar 2026

Conceptual Qualitative System Dynamics Model for Simulation of Perceived Workload, Stress and Performance from Industrial Work Content

PONE-D-25-57202R1

Dear Dr. Ruppert,

We’re pleased to inform you that your manuscript has been judged scientifically suitable for publication and will be formally accepted for publication once it meets all outstanding technical requirements.

An invoice will be generated when your article is formally accepted. Please note, if your institution has a publishing partnership with PLOS and your article meets the relevant criteria, all or part of your publication costs will be covered. Please make sure your user information is up-to-date by logging into Editorial Manager at Editorial Manager® and clicking the ‘Update My Information' link at the top of the page. For questions related to billing, please contact  and clicking the ‘Update My Information' link at the top of the page. For questions related to billing, please contact billing support..

Kind regards,

Gerard Hutchinson, MD

Academic Editor

PLOS One

Additional Editor Comments (optional):

Reviewers' comments:

Reviewer's Responses to Questions

**Comments to the Author**

1. If the authors have adequately addressed your comments raised in a previous round of review and you feel that this manuscript is now acceptable for publication, you may indicate that here to bypass the “Comments to the Author” section, enter your conflict of interest statement in the “Confidential to Editor” section, and submit your "Accept" recommendation.

Reviewer #1: All comments have been addressed

Reviewer #2: All comments have been addressed

2. Is the manuscript technically sound, and do the data support the conclusions?

Reviewer #1: Yes

Reviewer #2: Yes

3. Has the statistical analysis been performed appropriately and rigorously? 

Reviewer #1: Yes

Reviewer #2: Yes

4. Have the authors made all data underlying the findings in their manuscript fully available?

Reviewer #1: Yes

Reviewer #2: Yes

5. Is the manuscript presented in an intelligible fashion and written in standard English?

Reviewer #1: Yes

Reviewer #2: Yes

6. Review Comments to the Author

Reviewer #1: All the issues were addressed. Therefore, I do not have any other concerns and I recommend the manuscript for final publication.

Reviewer #2: The paper proposes a conceptual qualitative system dynamics model to simulate the stress and performance of workers in a given work environment and conditions.

The paper addresses a very interesting and appealing research topic; the approach and the topics discussed in the article are new and justify the interest in the publication. The structure of the paper is correct. The revisions adopted have improved the work. The suggestions proposed in the review report are included in the last draft of the paper. The major gaps filled by the manuscript are now described in “Model structure and simulated scenarios ”, and “Results”. The mistakes identified in the previous draft of the paper have been corrected.

Well done!

7. PLOS authors have the option to publish the peer review history of their article (what does this mean?). If published, this will include your full peer review and any attached files.). If published, this will include your full peer review and any attached files.

.

Reviewer #1: No

Reviewer #2: **Yes:** Francesco FacchiniFrancesco Facchini

---

## [Editor Report · Acceptance letter]

PONE-D-25-57202R1

PLOS One

Dear Dr. Ruppert,

I'm pleased to inform you that your manuscript has been deemed suitable for publication in PLOS One. Congratulations! Your manuscript is now being handed over to our production team.

Kind regards,

on behalf of

Dr. Gerard Hutchinson

Academic Editor

PLOS One